# Goal-conditioned Offline Planning from Curious Exploration

**Marco Bagatella**
ETH Zürich & Max Planck Institute for Intelligent Systems
Tübingen, Germany
mbagatella@ethz.ch

**Georg Martius**
University of Tübingen & Max Planck Institute for Intelligent Systems
Tübingen, Germany
georg.martius@uni-tuebingen.de

## Abstract

Curiosity has established itself as a powerful exploration strategy in deep reinforcement learning. Notably, leveraging expected future novelty as intrinsic motivation has been shown to efficiently generate exploratory trajectories, as well as a robust dynamics model. We consider the challenge of extracting goal-conditioned behavior from the products of such unsupervised exploration techniques, without any additional environment interaction. We find that conventional goal-conditioned reinforcement learning approaches for extracting a value function and policy fall short in this difficult offline setting. By analyzing the geometry of optimal goal-conditioned value functions, we relate this issue to a specific class of estimation artifacts in learned values. In order to mitigate their occurrence, we propose to combine model-based planning over learned value landscapes with a graph-based value aggregation scheme. We show how this combination can correct both local and global artifacts, obtaining significant improvements in zero-shot goal-reaching performance across diverse simulated environments.

## 1 Introduction

While the standard paradigm in reinforcement learning (RL) is aimed at maximizing a specific reward signal, the focus of the field has gradually shifted towards the pursuit of generally capable agents [1, 20, 40]. This work focuses on a promising framework for learning diverse behaviors with minimal supervision, by decomposing the problem into two consecutive phases. The first phase, referred to as *exploratory* [28, 43] or *intrinsic* [38], consists of unsupervised exploration of an environment, aiming to collect diverse experiences. These can be leveraged during the second phase, called *exploitatory* or *extrinsic*, to distill a family of specific behaviors without further environment interaction. This paradigm unifies several existing frameworks, including zero-shot deep reinforcement learning [28, 43], Explore2Offline [23], ExORL [52] and model-based zero-shot planning [38].

An established technique for the *intrinsic* phase is curiosity-based exploration, which leverages some formulation of surprise as a self-supervised reward signal. In practice, this is often done by estimating the uncertainty of forward dynamic predictions of a learned model [49]. At the end of the exploratory phase, this method produces a sequence of collected trajectories, as well as a trained dynamics model. At its core, our work focuses on the subsequent *extrinsic phase*, and is concerned with extracting strong goal-conditioned behavior from these two products, without further access to the environment.

37th Conference on Neural Information Processing Systems (NeurIPS 2023).

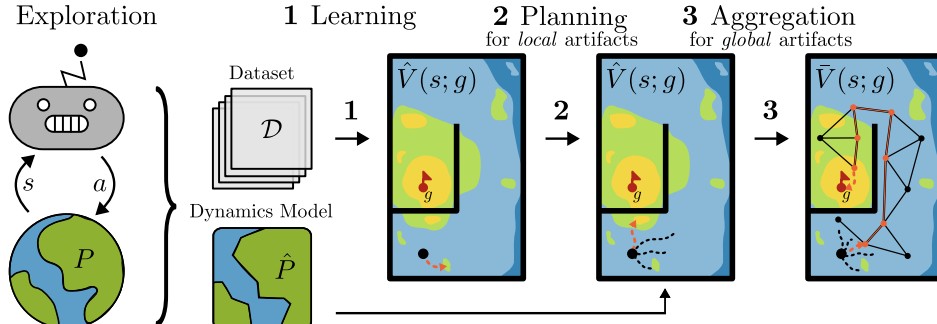

Figure 1: This work focuses on extracting goal-conditioned behavior from exploratory trajectories $\mathcal{D}$ and a trained dynamics model $\hat{P}$. We propose to *learn* a value function through goal-conditioned RL, *plan* with zero-order trajectory optimization to mitigate local estimation artifacts, and *aggregate* value estimates along a graph in order to correct global estimation artifacts.

In its simplest form, this corresponds to learning to control the agent's future state distribution in order to achieve a given goal or, more concretely, to visit a particular state. Existing methods tackle this problem without additional learning through model-based online planning [38], or rely on learning an amortized policy, either during exploration [28, 43] or in a fully offline manner [23, 52]. The former class of approaches requires the definition of a hand-crafted cost function, without which long-horizon, sparse tasks would remain out of reach. On the other hand, the latter class does not rely on a well-shaped cost function, but may suffer from myopic behavior in the absence of explicit planning, and from estimation errors in the learned value function.

As we do not assume access to external task-specific cost functions, we first investigate the effectiveness of goal-conditioned RL algorithms for learning a value function. In doing so, we empirically extend the results obtained by Yarats et al. [52] to previously unstudied model-based and goal-conditioned settings. We thus confirm that RL methods not involving offline corrections can be competitive when trained on offline exploratory data. Additionally, by a formal analysis of the geometry of optimal goal-conditioned value functions, we pinpoint a geometric feature of inaccurate value landscapes, which we refer to as $\mathcal{T}$-local optima. After showing the detrimental effects of such learning artifacts to the performance of goal-conditioned RL, we propose to mitigate them through a novel combination of model-based planning with a practical graph-based aggregation scheme for learned value estimates. Crucially, we show how these two components address the same estimation issue on different scales, that is, respectively, locally and globally. We provide a graphical overview in Figure 1 for a simple maze environment.

The contributions of this work can be organized as follows:

- We provide an empirical evaluation of goal-conditioned reinforcement learning algorithms when trained on the outcomes of an unsupervised exploration phase.
- We pinpoint a class of estimation artifacts in learned goal-conditioned value functions, which explains the suboptimality of naive, model-free reinforcement learning approaches in this setting.
- We propose to combine model-based planning with graph-based value aggregation to address the occurrence of both local and global value artifacts.
- We provide the first fully offline method for achieving goals after a curious exploration phase.

After the relating our work to existing literature, we evaluate goal-conditioned RL methods when learning from curious exploration (Section 4). We then pinpoint a class of estimation artifacts that are not compatible with optimal goal-conditioned value functions and analyze them through the lens of $\mathcal{T}$-local optima (Section 5). Equipped with these insights, we show how our practical method can mitigate such artifacts (Section 6). A compact description of the resulting method, highlighting its intrinsic and extrinsic phases, can be found in Appendix H.

## 2 Related Work

**Unsupervised Exploration** The challenge of exploration has been at the core of reinforcement learning research since its inception. Within deep reinforcement learning, successful strategies are

often derived from count-based [2], prediction-based [41] or direct exploration techniques [10], with the first two traditionally relying on intrinsic motivation [7] computed from state visitation distributions or prediction errors (i.e., *surprise*), respectively. Depending on the entity on which prediction error is computed, methods can focus on estimating forward dynamics [32, 38, 43, 44], inverse dynamics [33] or possibly random features of the state space [3]. Another important distinction falls between methods that retroactively compute surprise [33] or methods that compute future expected disagreement [38, 43], and proactively seek out uncertain areas in the state space. Overall, such techniques compute an additional intrinsic reward signal, which can be summed to the extrinsic environment reward signal, or used alone for fully unsupervised exploration. In this work, we build upon efficient exploration methods that are proactive, and rely on prediction disagreement of a forward dynamics model [38, 43, 44]. Nevertheless, our method could be easily applicable in conjunction with any aforementioned strategy, as long as it can produce exploratory data, and a dynamics model, which could potentially also be learned offline.

**Zero-shot Frameworks**   Unsupervised exploration techniques are capable of efficiently sampling diverse trajectories, without extrinsic reward signals. As a result, they are a promising source of data to learn flexible behaviors. Several works in the literature have proposed that a task-agnostic unsupervised exploration phase be followed by a second phase, optimizing a task-specific reward function only provided a posteriori. This has been formalized as zero-shot reinforcement learning [9, 47, 48]. Notably, this framework conventionally forbids task-specific fine-tuning, explicit planning or computation during the second phase. On the applied side, agents endowed with learned world models [28, 43] have shown promise for zero-shot generalization without additional learning or planning. Other works are less strict in their constraints: Sancaktar et al. [38] also propose an intrinsic exploration phase followed by zero-shot generalization, but the latter relies on task-specific online planning. Furthermore, two concurrent works [23, 52] study the application of offline deep reinforcement learning for the second phase. However, these methods are not designed for goal-conditioned tasks, and thus require an external task-specific reward function to perform relabeling during training. Our framework fits within this general research direction, in that it also involves task-agnostic and task-specific components. The initial phase is task-agnostic: we rely on unsupervised exploration, and train a universal value estimator [40] completely offline, without access to an external task-specific reward signal. In the second phase, we perform model-based planning via zero-order trajectory optimization to control the agent. We remark that, while this requires significant computation, existing algorithms are fast enough to allow real-time inference [35].

**Goal-Conditioned Reinforcement Learning**   The pursuit of generally capable agents has been a ubiquitous effort in artificial intelligence research [36, 42, 45]. Goal-conditioned RL represents a practical framework for solving multiple tasks, by describing each task through a goal embedding, which can then condition reward and value functions. Deep RL methods can then be used to estimate values by providing the goal as an additional input to their (often neural) function approximators [40]. A ubiquitous technique in these settings is that of goal relabeling [1], which substitutes a trajectory's goal with one sampled a posteriori. Goal-conditioned methods have been shown to be successful in the offline setting [46], often when coupled with negative sampling methods [5], but have been mostly evaluated on standard offline datasets. An alternative line of work frames the problem as supervised learning instead [11], and shows connections to the original reinforcement learning objective [51]. Additionally, several works have explored the combination of goal-conditioned learning with model-based planning [4, 25, 31]. This paper studies the natural application of goal-conditioned RL after an unsupervised exploration phase, which allows a straightforward definition of the multiple tasks to be learned. Moreover, our main formal insights in Section 5 build on top of common assumptions in goal-conditioned RL.

**Graph-based Value Estimation**   Several existing works rely on graph abstractions in order to retrieve global structure and address value estimation issues. In particular, Eysenbach et al. [12] and Savinov et al. [39] propose high-level subgoal selection procedures based on shortest paths between states sampled from the replay buffer. Another promising approach proposes to build a graph during training, and to leverage its structure when distilling a policy [30]. In contrast, our method relies on discrete structures only at inference, and directly integrates these structures into model-based planning, thus removing the need for subgoal selection routines.

# 3 Preliminaries

## 3.1 Background and Notation

The environments of interest can be modeled as Markov decision processes (MDP) $\mathcal{M} = \{\mathcal{S}, \mathcal{A}, P, R, \mu_0, \gamma\}$, where $\mathcal{S}$ and $\mathcal{A}$ are (possibly continuous) state and action spaces, $P : \mathcal{S} \times \mathcal{A} \to \Omega(\mathcal{S})$ is the transition probability distribution [1], $R : \mathcal{S} \times \mathcal{A} \to \mathbb{R}$ is a reward function, $\mu_0 \in \Omega(\mathcal{S})$ is an initial state distribution, and $\gamma$ is a discount factor.

Goal-conditioned reinforcement learning augments this conventional formulation by sampling a goal $g \sim \mu_{\mathcal{G}}$, where $p_{\mathcal{G}}$ is a distribution over a goal space $\mathcal{G}$. While goal abstraction is possible, for simplicity of notation we consider $\mathcal{G} \subseteq \mathcal{S}$ (i.e., goal-conditioned behavior corresponds to reaching given states). The reward function is then defined as $R(s, a; g) = R(s; g) = \mathbf{1}_{d(s,g) \leq \epsilon}$ for some $\epsilon \geq 0$, and under some distance metric $d(\cdot)$. We additionally assume that episodes terminate when a goal is achieved, and the maximum undiscounted return within an episode is thus one. It is now possible to define a goal-conditioned policy $\pi : \mathcal{S} \times \mathcal{G} \to \Omega(\mathcal{A})$ and its value function $V^\pi(s_0; g) = \mathbb{E}_{P,\pi} \sum_{t=0}^{\infty} \gamma^t R(s_t, g)$, with $s_t \sim P(\cdot|s_{t-1}, a_{t-1})$, $a_t \sim \pi(s_t; g)$. The objective of a goal-conditioned RL agent can then be expressed as finding a stochastic policy $\pi^\star = \text{argmax}_\pi \mathbb{E}_{g \sim p_{\mathcal{G}}, s \sim p_0} V^\pi(s; g)$.

When learning from curious exploration, we additionally assume the availability of a set $\mathcal{D} = \{(s_0, a_0, s_1, a_1, \ldots, s_{T_i})_i\}_{i=1}^{N}$ of $N$ exploratory state-action trajectories of possibly different lengths $T_i$, as well as access to an estimate of the probabilistic transition function $\hat{P} \approx P$, i.e., a dynamics model.

## 3.2 Data Collection and Experimental Framework

As our contributions are motivated by empirical observations, we also introduce our experimental pipeline early on. Algorithms are evaluated in four MDP instantiations, namely `maze_large` and `kitchen` from D4RL [14], `fetch_push` from gymnasium-robotics [13] and an adapted `pinpad` environment [18]. These environments were chosen to drastically differ in dynamics, dimensionality, and task duration. In particular, `maze_large` involves the navigation to distant goals in a 2D maze by controlling the acceleration of a point mass, `kitchen` requires controlling a 9 DOF robot through diverse tasks, including manipulating a kettle and operating switches, `fetch_push` is a manipulation task controlled through operation space control, and `pinpad` requires navigating a room and pressing a sequence of buttons in the right order.

For each environment, we collect 200k exploratory transitions by curious exploration, corresponding to ~2 hours of real-time interaction at 30Hz, which is an order of magnitude less than existing benchmarks [14, 24]. We stress that this amount of data is realistically collectible in the real world with minimal supervision, thus providing a strong benchmark for data-efficient learning of general agents. Moreover, the nature of the data is significantly different from standard offline datasets, which often contain high-quality demonstrations [14] that can be distilled into policies through naive imitation.

The environment's dynamics are estimated through an ensemble of stochastic neural networks, similar to the one proposed in Chua et al. [8]. The intrinsic exploration signal is ensemble disagreement [38, 43, 49], as described in Appendix I.2. All algorithms are evaluated by success rate, that is, the ratio of test episodes in which the commanded goal is achieved. Details can be found in Appendix I.

# 4 Value Learning from Curious Exploration

Curious exploration produces a learned dynamics model $\hat{\mathcal{P}}$ and a set $\mathcal{D}$ of trajectories, lacking both goal and reward labels. In order to achieve arbitrary (and potentially distant) goals at inference time, and considering that a task-specific cost function is not externally provided, it is natural to resort to learning and optimizing a value function. This section evaluates established methods for value learning and studies the quality of their estimates, thus motivating the analysis and method proposed in Sections 5 and 6.

---

[1]We use $\Omega(\mathcal{S})$ to represent the space of probability distributions over $\mathcal{S}$.

A standard approach to train a goal-conditioned value function from unlabeled data involves two steps. The first step consists of a goal-relabeling procedure, which associates each training sample with a goal selected a posteriori, and computes a self-supervised reward signal from it, as briefly explained in Section 4.1. The second step consists of extracting a goal-conditioned value function by applying off-policy RL algorithms on the relabeled data, as analyzed in Section 4.2. While this work focuses on data collected through curious exploration, the methods described can also be applied to a general offline goal-conditioned reinforcement learning settings, for which we provide a minimal study in Appendix F.

## 4.1 Relabeling Curious Exploration through Self-supervised Temporal Distances

We follow the framework proposed by Tian et al. [46] to relabel exploration trajectories, and compute self-supervised reward signals by leveraging temporal distances, without assuming access to external task-specific reward functions. Training samples $(s_t, a_t, s_{t+1})$ are sampled uniformly from the replay buffer $\mathcal{D}$, and need to be assigned a goal $g$ and a reward scalar $r$. Each training sample is assigned a *positive* goal with probability $p_g$, and a *negative* goal otherwise. Positive goals are sampled as future states from the same trajectory: $g = s_{t+\tau}$, where $\tau \sim \text{Geo}(p_{\text{geo}})$ follows a geometric distribution with parameter $p_{\text{geo}}$. Negative goals are instead sampled uniformly from the entire dataset, while Tian et al. [46] rely on prior knowledge on the semantics of the state space to extract challenging goals. Rewards for all state-goal pairs are simply computed using the reward function $r = R(s_{t+1}; g) = \mathbf{1}_{s_{t+1}=g}$: the reward is non-zero only for positive goals sampled with $\tau = 1$. This corresponds to a minorizer for the class of reward functions $R(s_{t+1}; g) = \mathbf{1}_{d(s_{t+1},g) \leq \epsilon}$, as setting $\epsilon = 0$ effectively removes the necessity to define a specific distance function $d$ over state and goal spaces. We remark that while its extreme sparsity prevents its optimization via zero-shot model based planning, it remains functional as a reward signal for training a value function in hindsight.

## 4.2 Value Estimation and Suboptimality

Now that sampling and relabeling procedures have been defined, we focus our attention on which methods can effectively retrieve a goal-conditioned value function $V(s; g)$. In particular, we evaluate off-policy actor-critic algorithms, which are in principle capable of extracting the optimal value function (and policy) from trajectories generated by a suboptimal policy (as the one used for exploration). We thus ask the following questions:

1. *Which off-policy actor-critic algorithm is most suitable for learning a goal-conditioned value function, purely offline, from data collected through unsupervised exploration?*

2. *How close to optimality is the learned value function?*

To answer the first question, we consider several off-policy actor-critic algorithms, and evaluate the learned value function through the success rate of an agent which seeks to optimize it. The choice of algorithms should adhere to the offline setting, as well as the availability of an estimated dynamics model $\hat{P}$. We thus evaluate TD3 [15] as default model-free actor-critic algorithm, and

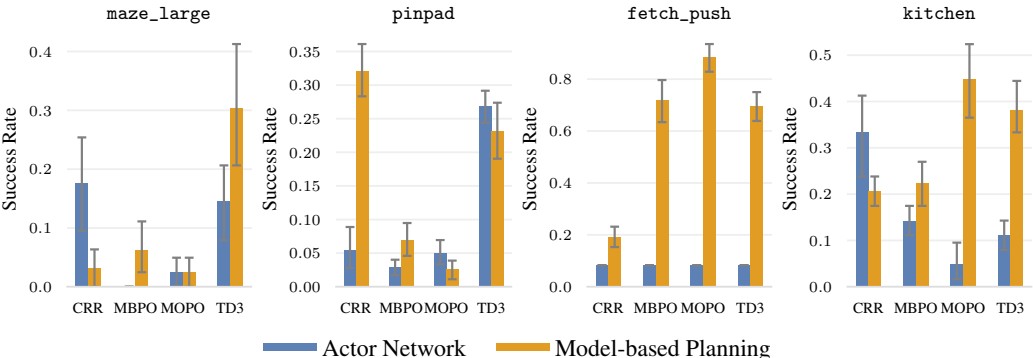

Figure 2: Performance of actor networks (blue) and model-based planning (orange) on value functions trained through various actor-critic algorithms from the same curious exploration datasets.

compare its performance to CRR [50], a state-of-the-art offline method [16]. We also evaluate two model-based methods, namely MBPO [19] and MOPO [53], the latter of which introduces an uncertainty-based penalty for offline learning. Further value-learning baselines are presented in Appendix C. All methods have access to the same offline exploration dataset $\mathcal{D}$; model-based method additionally query the dynamics model $\hat{\mathcal{P}}$ in order to generate training trajectories. Algorithm-specific hyperparameters were tuned separately for each method through grid search, and kept constant across environments. Each algorithm is allowed the same number of gradient steps (sufficient to ensure convergence for all methods). We implement two strategies for optimizing the value functions: querying the jointly learned actor network, and employing model-based planning and using zero-order trajectory optimization (e.g., CEM [37]/iCEM[35]) with the learned model $\hat{P}$ to optimize $n$-step look ahead value estimates [46]. Average success rates over 9 seeds are presented with 90% simple bootstrap confidence intervals in Figure 2.

To answer the first question, we consider the performance of the best optimizer (often model-based planning, in orange) for each of the value learning methods. We observe that simpler actor-critic algorithms (i.e., TD3) are overall competitive with the remaining value learning methods. MBPO, MOPO and CRR are not consistently better in the considered environments, despite involving offline corrections or relying on the learned model $\hat{P}$ to generate training trajectories. We hypothesize that this phenomenon can be attributed, at least partially, to the exploratory nature of the data distribution, as exemplified in Figure 3. Additionally, we note that our findings and hypothesis are consistent with those reported by Yarats et al. [52] for non-goal-conditioned tasks.

We then turn to examining the performance gap between actor networks and model-based planning. We observe a tendency for model-based planning to outperform actor networks, with the largest improvements in fetch_push, as all learned policies similarly fail in this environment. While this stands true for almost all considered algorithms, CRR poses an exception to the success of model-based planning for two environments, which we hypothesize is due to its reliance on weighted behavior cloning rather than on backpropagation through a learned critic.

We remark that it is possible to show that, under common assumptions, a policy which exactly optimizes an optimal goal-conditioned value function is guaranteed to achieve its goals (see Appendix A.2). The low success rate of actor networks, assuming their objective is well-optimized, is thus a symptom of suboptimality for learned value functions. In Section 5, we argue how this suboptimality can be partially traced back to a class of estimation artifacts; moreover, we show how model-based planning is intrinsically more robust to such artifacts than actor networks (see Section 6.1).

Due to its strong empirical performance, TD3 is adopted for value learning through the rest of this work. We however remark that the analysis and the method proposed in Sections 5 and 6 remain valid for each of the RL methods considered. Further details on methods and results presented in this section are reported in Appendix I.

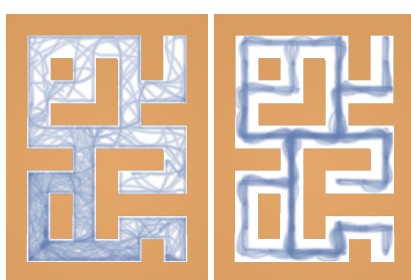

Figure 3: 200k transitions from the exploration dataset $\mathcal{D}$ (left), and from the standard offline dataset provided by D4RL (right). Unsupervised exploration leads to a more uniform state coverage, and eases out-of-distribution issues.

## 5 Pathologies of Goal-Conditioned Value Functions

The performance gap between model-based planning and actor networks described in the previous section is a symptom of estimation issues in learned value functions. In this section, we pinpoint a particular class of geometric features which are not compatible with optimal goal-conditioned value functions. Such features represent a significant failure case for an agent optimizing a learned value function, as they can effectively trap it within a manifold of the state space. Intuitively, this happens when the estimated value of the current state exceeds that of all other reachable states, despite the goal not being achieved.

We now formalize this intuition by defining such geometric features as $\mathcal{T}$-local optima. We then show that $\mathcal{T}$-local optima are not present in optimal goal-conditioned value functions, and are therefore

estimation artifacts. This justifies the method proposed in Section 6, which is designed to mitigate this class of artifacts.

Let us define the set operator $\mathcal{T} : \mathbb{P}(\mathcal{S}) \rightarrow \mathbb{P}(\mathcal{S})$ [2] that returns the set of states reachable in one step from a given initial set of states $\mathcal{S}_0 \subseteq \mathcal{S}$:

$$\mathcal{T}(\mathcal{S}_0) = \bigcup_{s_o \in \mathcal{S}_0} \{s' \in \mathcal{S} \mid \exists\, a \in \mathcal{A} : P(s'|s_0, a) > 0\}. \tag{1}$$

We can now formally introduce a class of geometric features, which we refer to as $\mathcal{T}$-local optima:

**Definition 5.1.** A state $s^* \in \mathcal{S}$ is a $\mathcal{T}$-local optimum point for a value function $V(s)$ if and only if

$$\max_{s' \in \mathcal{S}} V(s') > V(s^*) > \max_{s' \in \mathcal{T}(\{s^*\}) \setminus \{s^*\}} V(s').$$

Intuitively, a $\mathcal{T}$-local optimum is a *hill* in the value landscape over the state space, where proximity is defined by the transition function of the MDP. The first inequality ensures that $s^*$ does not maximize the value function globally. The second inequality guarantees that, if there is an action $\bar{a} \in \mathcal{A} : P(s^*|s^*, a) = 1$, this action would be chosen by greedily optimizing the value of the next-state (i.e., $\bar{a} = \mathrm{argmax}_{a \in \mathcal{A}} \int_{s' \in \mathcal{S}} V(s') P(s'|s^*, a)$), thus trapping the agent in $s^*$. By extending the previous definition to the multistep case, we can also conveniently define *depth* as the minimum number of steps required to reach a state with a higher value (i.e., *escape* the $\mathcal{T}$-local optimum).

**Definition 5.2.** A $\mathcal{T}$-local optimum point $s^*$ has depth $k \geq 1$ if $k$ is the smallest integer such that

$$V(s^*) < \max_{s' \in \mathcal{T}^{k+1}(\{s*\})} V(s').$$

Intuitively, a $\mathcal{T}$-local optimum of depth $k$ dominates the value of states reachable in up to $k$ steps.

These definitions prepare the introduction of a core observation. Within the setting of goal-conditioned RL, $\mathcal{T}$-local optima are not an intrinsic feature of the value landscape, but rather an estimation artifact.

**Proposition 5.3.** *Given a goal $g \in \mathcal{G}$, an optimal goal-conditioned value function $V^*(s; g)$ has no $\mathcal{T}$-local optima.*

The proof follows directly from Bellman optimality of the value function under standard assumptions of goal-conditioned RL introduced in Section 3.1. We report it in its entirety in Appendix A.1.

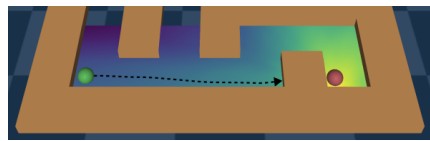

Figure 4: Learned value as a function of $x$-$y$ coordinates in `maze_large`. Choosing actions that maximize the value of the next state traps the agent in a corner, i.e. a $\mathcal{T}$-local optimum point.

We remark that, in principle, showing that a state $s$ is a $\mathcal{T}$-local optimum point requires evaluating the value function over the support of the next state distribution, which would require access to the true dynamics of the MDP. Moreover, exhaustive evaluation is unfeasible in continuous state spaces. However, one can bypass these issues by instead *estimating*, rather than proving, the presence of $\mathcal{T}$-local optima. Given a candidate $\mathcal{T}$-local optimum point $s$, $V(\cdot; g)$ can be evaluated on (i) random states $s_r$ sampled from the exploration data $\mathcal{D}$ instead of the state space $\mathcal{S}$ and (ii) samples $s'$ from the next state distribution, accessed through the dynamic model $\hat{P}(s'|s, a)$ with $a \sim \mathcal{U}(\mathcal{A})$. The maximum operators in Definition 5.1 can then be estimated through sampling; if both inequalities hold, $s$ can be estimated to be a $\mathcal{T}$-local optimum point.

Having defined $\mathcal{T}$-local optima, and shown that they are learning artifacts, we now investigate their impact on goal-reaching performance. We train $n = 30$ bootstrapped value estimators for all environments, and for each of them we generate trajectories through model-based planning with horizon $H$. We then estimate the occurrence of $\mathcal{T}$-local optima of depth $k \geq H$ by testing each state along each trajectory, as described above. This is reported in Figure 5, separately for successful and unsuccessful trajectories. We observe that, in most environments, trajectories which avoid $\mathcal{T}$-local optima generally tend to be more likely to succeed; this tendency tends to fade in environments which only require short-term planning (e.g., `fetch_push`). We note that these estimates are sampled-based, and thus noisy by nature, especially in low-data regimes; we thus expand on this analysis in Appendix B. For illustrative purposes, we also visualize a $\mathcal{T}$-local optimum in the value landscape over `maze_large` in Figure 4.

---

[2]$\mathbb{P}(\cdot)$ represents a power set.

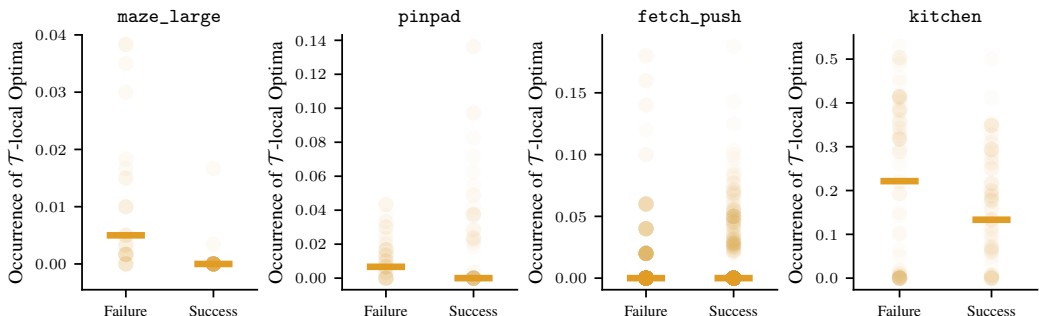

Figure 5: Occurrence of $\mathcal{T}$-local optima of depth $k \geq H$ along successful and unsuccessful trajectories generated through model-based planning with horizon $H = 15$, estimated via sampling. An horizontal bar marks the median.

# 6 Mitigating Local and Global Value Artifacts

Having shown that $\mathcal{T}$-local optima are estimation artifacts, we now focus on categorizing and mitigating their occurrence. A first observation, which confirms the results in Section 4, is that model-based planning is robust to $\mathcal{T}$−local optima of small depth. This is further argued in Section 6.1. However, this does not address the occurrence of global estimation artifacts (i.e. $\mathcal{T}$-local optima of large depth). We thus propose a graph-based value aggregation scheme for this purpose in Section 6.2. Finally, in Section 6.3 we empirically show how combining these two components can address estimation artifacts both locally and globally, and consistently improves goal-reaching performance.

## 6.1 Model-based Planning for Local Value Artifacts

We will now provide further arguments for the effectiveness of model-based planning over inaccurate value landscapes, supported by Figure 2. Model-based planning is a model-predictive control (MPC) algorithm, which finds an action sequence $(a_0, \ldots a_{H-1})$ by optimizing the RL objective through a $H$-step value estimate for a given horizon $H$, dynamics model $\hat{P}$, state $s_0$, and goal $g$:

$$(a_0, \cdots, a_{H-1}) = \operatorname*{argmax}_{(a_0, \cdots, a_{H-1}) \in \mathcal{A}^H} \sum_{i=0}^{H-1} \gamma^i R(s_i, g) + \gamma^H V(s_H; g) \quad \text{with } s_{i+1} \sim \hat{P}(\cdot|s_i, a_i), \ (2)$$

and executes the first action $a_0$. We remark that in our setting, due to the extreme sparsity of $R(s; g) = \mathbf{1}_{s=g}$ in continuous spaces, the objective in Equation 2 simplifies to the maximization of $V(s_H; g)$ alone. For the same reason, it is easy to show that an actor network trained through policy gradient is optimizing the same objective as model-based planning with horizon $H = 1$:

$$a_0 = \operatorname*{argmax}_{a_0 \in \mathcal{A}} Q(s_0, a_0; g) = \operatorname*{argmax}_{a_0 \in \mathcal{A}} R(s_0; g) + \gamma V(s_1; g) = \operatorname*{argmax}_{a_0 \in \mathcal{A}} V(s_1; g), \quad (3)$$

with $s_1 \sim P(\cdot|s_0, a_0)$. Under certain assumptions, such as exact dynamics estimation (i.e., $\hat{P} = P$), exact optimization for Equation 2 and existence of self-loops in the MDP, it is possible to show that (i) model-based planning from a $\mathcal{T}$-local optimum point of depth $k$ is guaranteed to fail if $H \leq k$ and (ii) model-based planning is guaranteed to reach a state with a higher value if the horizon $H$ is greater than the depth of the largest $\mathcal{T}$-local optima over the value landscape. For a complete argument, see Appendix A.3. As a consequence, actor networks can be trapped in *all* local optima, while model-based planning with horizon $H > 1$ can potentially escape local optima of depth $k < H$, and is therefore potentially more robust to such estimation artifacts, as shown empirically in Figure 2.

## 6.2 Graph-based Aggregation for Global Value Artifacts

While we have shown that model-based planning is robust to local artifacts, the correction of global estimation artifacts, which span beyond the effective planning horizon $H$, remains a significant challenge. We propose to address it by *grounding* the value function in states observed during exploration, such that long-horizon value estimates are computed by aggregating short-horizon (i.e.,

local) estimates. Given the current state $s \in \mathcal{S}$ and the goal $g \in \mathcal{G}$, we build a sparse graph by sampling vertices from the empirical state distribution $\mathcal{D}_s$ from the replay buffer, and connecting them with edges weighted by the learned goal-conditioned value $\hat{V}$:

$$\mathbf{G} = (\mathcal{V}, \mathcal{E}) \text{ with } \mathcal{V} \subset \mathcal{D}_s \cup \{s, g\} \text{ and } \mathcal{E} = \{(s', s'', \hat{V}(s'; s'')) \, \forall \, (s', s'') \in \mathcal{V} \times \mathcal{V}\}. \quad (4)$$

Crucially, we then prune the graph by removing all edges that encode long-horizon estimates, i.e., edges whose weight is below a threshold $V_{\min}$. In practice, we set $V_{\min}$ to the maximum value which still ensures connectivity between the agent's state $s$ and the goal $g$ along the graph. This can be computed efficiently, as detailed in Appendix D. Once the graph has been constructed and pruned, we simply compute value estimates by minimizing the product of value estimates along all paths to the goal.

$$\bar{V}(s; g) = \min_{(s_0, \ldots, s_T)} \prod_{i=0}^{T-1} \hat{V}(s_i; s_{i+1}) \text{ where } (s_0, \ldots, s_T) \text{ is a path between } s \text{ and } g \text{ on } \mathbf{G} \quad (5)$$

This value aggregation scheme differentiates itself from existing methods [12] in its vertex sampling phase, which relies on density estimation to address skewed state distributions, and in its pruning phase, which does not depend on hyperparameters. Crucially, the final value aggregate can be directly adapted as a cost function for model-based planning, and is not used as a proxy for subgoal selection. These differences and their consequences are described extensively in Appendix D.

In practice, this scheme can mitigate global $\mathcal{T}$-local optima by replacing long-horizon value estimates through an aggregation of short-horizon estimates. Such short-horizon estimates have been shown to be generally more accurate, and useful in containing value overestimation issues [12], which can be considered a generalization of $\mathcal{T}$-local optima.

For reference, we now provide a visual intuition for the effect of the graph-based value aggregation scheme we propose. We highlight this in `maze_large`, as it is prone to global estimation artifacts, and allows straightforward visualization. The blue column in Figure 6 plots value functions $\{\hat{V}(s; g_i)\}_{i=0}^2$ learned through TD3 against $x$-$y$ coordinates for three different goals. The green column reproduces the same plots, but estimates values through graph-based value aggregation, as described in this Section, resulting in corrected estimates $\{\bar{V}(s; g_i)\}_{i=0}^2$. We further compute expert trajectories for all three goals, in red, and plot $\hat{V}$ and $\bar{V}$ along them, resulting in the three plots on the right. When comparing value estimates with optimal values ($V^*$, in orange, computed in closed form), we observe that graph-based aggregation results in more accurate estimates. Crucially, it is able to address global estimation artifacts, which result in a "bleeding" effect for $\hat{V}(s; g_0)$ (on the left), or in large "hills" in $\hat{V}(s_t; g_0)$ (on the right, in blue).

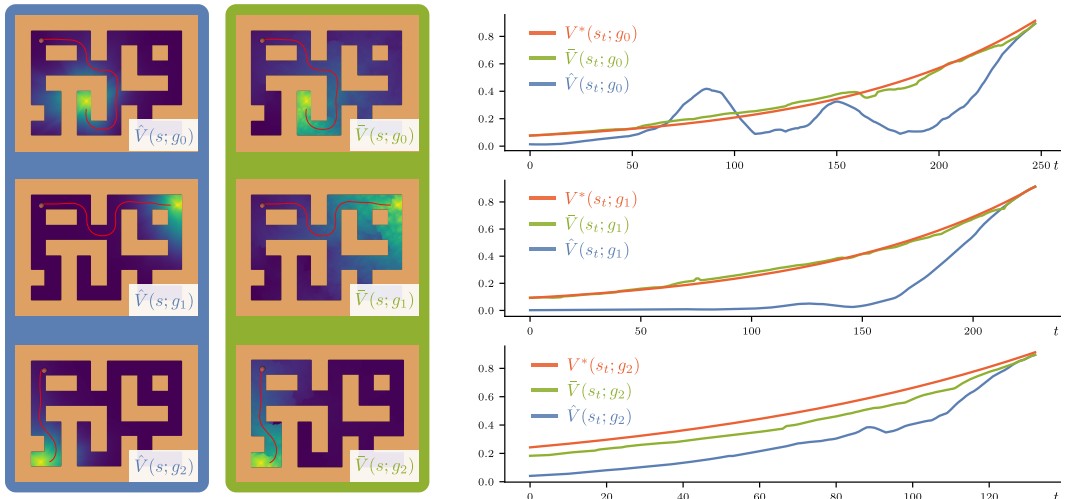

Figure 6: Learned values $\hat{V}(s; g_i)$ as a function of $x$-$y$ coordinates in `maze_large` (blue column). Graph-based value aggregation results in corrected estimates $\bar{V}(s; g_i)$ (green column). Values along optimal trajectories are compared to optimal value functions $V^*(s_t; g_i)$ on the right.

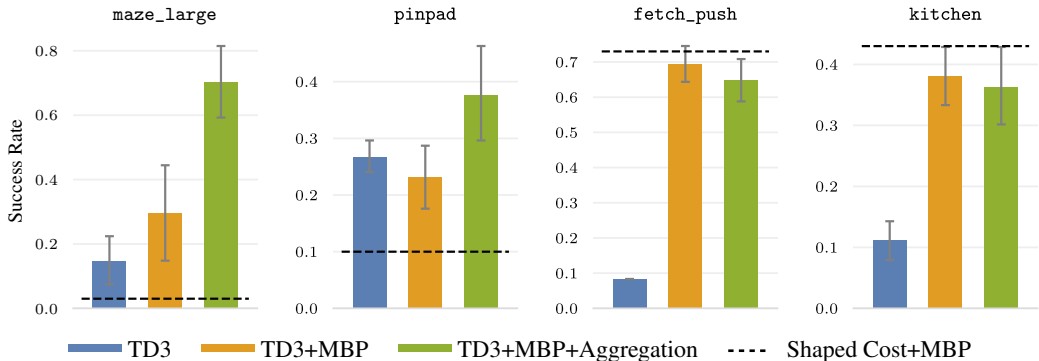

Figure 7: Performance of an actor network (TD3) against model-based planning without (TD3+MBP) and with graph-based value aggregation (TD3+MBP+Aggregation). Model-based planning on aggregated values addresses both local and global estimation artifacts, and performs well across diverse environments.

## 6.3 Experimental Evaluation

Having evaluated diverse value learning algorithms, and described a class of harmful estimation artifacts, we now set out to evaluate how such artifacts can be addressed both locally and globally. Figure 7 compares the performance of actor networks with that of model-based planning, with and without graph-based value aggregation. We observe that model-based planning improves performance in most tasks, as it addresses $\mathcal{T}$-local optima of limited depth. However, it fails to correct global artifacts in long-horizon tasks (i.e., `maze_large` and `pinpad`). In this case, we observe that graph-based value aggregation increases success rates significantly. Furthermore, we observe that our method, which combines these two components, consistently achieves strong performance across diverse environments.

In each plot, a dashed line represents the performance of model-based planning with an external sparse cost (for some $\epsilon > 0$). On one hand, its strong performance in environments with shorter optimal trajectories confirms the accuracy of the learned model. On the other, its failure for long-horizon tasks shows the limitation of planning in the absence of a learned value function, which has been an important motivation for this work.

## 7 Discussion

This work is concerned with the challenge of achieving goals after an unsupervised exploration phase, without any further interaction with the environment. In this challenging setting, we observe that simple actor-critic methods are surprisingly competitive with model-based, or offline algorithms, but their actor networks are often suboptimal due to estimation artifacts in the learned value function. After characterizing these artifacts by leveraging the goal-conditioned nature of the problem, we show how model-based planning can address this problem locally, graph-based value aggregation can apply global corrections, and their combination works consistently across diverse environments.

**Limitations** With respect to simply querying actor networks, this method comes at a cost. In particular, it increases computational requirements and risks introducing further artifacts due to model exploitation, or the discretization involved in graph-based aggregation (see Appendix D). Furthermore, the proposed method is only applied at inference and, while it's able to mitigate learning artifacts, it does not directly correct them by updating the value estimator. Nevertheless, within our settings, we find the impact of these limitations to be, in practice, modest.

Our method represents a strong practical approach for goal-reaching after curious exploration, and sheds light on interesting properties of both offline and goal-conditioned problems. An exciting future avenue would further leverage these formal insights, possibly through a tighter integration of value learning and value correction. We thus believe that further exploring the interplay between a theoretically sound analysis of the properties of goal-conditioned value function, and the design of practical algorithms, represents a rich research opportunity.

## Acknowledgments and Disclosure of Funding

We thank Andreas Krause, Nico Gürtler, Pierre Schumacher, Núria Armengol Urpí and Cansu Sancaktar for their help throughout the project, and the anonymous reviewers for their valuable feedback. Marco Bagatella is supported by the Max Planck ETH Center for Learning Systems. Georg Martius is a member of the Machine Learning Cluster of Excellence, EXC number 2064/1 – Project number 390727645. We acknowledge the support from the German Federal Ministry of Education and Research (BMBF) through the Tübingen AI Center (FKZ: 01IS18039B).

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

# Supplementary Material to
# Goal-conditioned Offline Planning from Curious Exploration

## A   Proofs

### A.1   Proposition 5.3

Let us consider an arbitrary goal $g \in \mathcal{G}$ and the optimal goal-conditioned value function $V^*(s; g)$. Proposition 5.3 states that $V^*(s; g)$ has no $\mathcal{T}$-local optima. The proof follows directly from the Bellman optimality condition, assuming that (1) $R(s; g) = \mathbf{1}_{d(s,g) \leq \epsilon}$ for some $\epsilon \geq 0$ and distance function $d(\cdot, \cdot)$, and that (2) the episode terminates when the goal is achieved.

For a candidate state $\tilde{s}$, let us assume that the first inequality of the condition for being a $\mathcal{T}$-local optimum point is met: $\max_{s' \in \mathcal{S}} V^*(s'; g) > V^*(\tilde{s}; g)$[3]. It is now sufficient to show that the second inequality (i.e., $V^*(\tilde{s}; g) > \max_{s' \in \mathcal{T}(\{\tilde{s}\}) \setminus \{\tilde{s}\}} V^*(s'; g)$) cannot hold, and therefore $\tilde{s}$ cannot be a $\mathcal{T}$-local optimum point. According to Bellman optimality, we have that

$$V^*(\tilde{s}; g) = R(\tilde{s}; g) + \gamma(\tilde{s}; g) \max_{a \in \mathcal{A}} \mathbb{E}_{s' \sim \mathcal{P}(\cdot|a, \tilde{s})} V^*(s'; g) \tag{S6}$$

where we use $\gamma(s; g) = \gamma(1 - R(s; g))$: $(1 - R(s; g))$ acts as the termination signal, to enforce that value propagation stops when reaching the goal (and the episode ends). If $R(\tilde{s}; g) = 1$, then $V^*(\tilde{s}; g) = 1$, which would break the first inequality, since $\max_{s' \in \mathcal{S}} V^*(s'; g) \leq 1$. This implies that $R(\tilde{s}; g) = 0$. As a consequence, Equation S6 simplifies to

$$V^*(\tilde{s}; g) = \gamma \max_{a \in \mathcal{A}} \mathbb{E}_{s' \sim \mathcal{P}(\cdot|a, \tilde{s})} V^*(s'; g)$$
$$\leq \gamma \max_{s' \in \mathcal{T}(\{\tilde{s}\})} V^*(s'; g) \tag{S7}$$

We can now differentiate two cases. If $\max_{s' \in \mathcal{T}(\{\tilde{s}\})} V^*(s'; g) = 0$ (i.e., the goal cannot be reached), then we also have that $V^*(\tilde{s}; g) = \max_{s' \in \mathcal{T}(\{\tilde{s}\}) \setminus \{\tilde{s}\}} V^*(s'; g) = 0$, and $\tilde{s}$ is not a $\mathcal{T}$-local optimum point.

If the goal is instead reachable from at least one state in the next state distribution (i.e., $\max_{s' \in \mathcal{T}(\{\tilde{s}\})} V^*(s'; g) > 0$), we have that

$$V^*(\tilde{s}; g) \leq \gamma(\tilde{s}; g) \max_{s' \in \mathcal{T}(\{\tilde{s}\})} V^*(s'; g)$$
$$< \max_{s' \in \mathcal{T}(\{\tilde{s}\})} V^*(s'; g)$$
$$= \max_{s' \in \mathcal{T}(\{\tilde{s}\}) \setminus \{\tilde{s}\}} V^*(s'; g). \tag{S8}$$

The second inequality follows from the goal-conditioned assumption that $0 < \gamma < 1$, and the final equality is due to the fact that $V^*(\tilde{s}; g) < \max_{s' \in \mathcal{T}(\{\tilde{s}\})} V^*(s'; g)$ implies that $V^*(\tilde{s}; g) \neq \max_{s' \in \mathcal{T}(\{\tilde{s}\})} V^*(s'; g)$. Thus, $V^*(\tilde{s}; g) > \max_{s' \in \mathcal{T}(\{\tilde{s}\}) \setminus \{\tilde{s}\}} V^*(s'; g)$ does not hold, and $\tilde{s}$ is not a $\mathcal{T}$-local optimum point.

### A.2   Optimization of Optimal Goal-conditioned Value Functions

Section 4 argues that, assuming that the objective for actor networks is well optimized, the failure of naive actor-critic algorithms is due to estimation errors in the critic network. Accordingly,

---

[3]While a supremum operator $\sup(\cdot)$ would be required in open domains, for simplicity we use $\max(\cdot)$ throughout this work.

we show that, under common assumptions, a state-space trajectory which optimizes an optimal goal-conditioned value function $V^*(s; g)$ is guaranteed to achieve its goal $g \in \mathcal{G}$. Furthermore, minimization of the actor loss results in an actor network which produces this exact trajectory.

Given a goal $g \in \mathcal{G}$, and an initial state $s_0 \in \mathcal{S}$, we make the following assumptions [4]:

1. $V^*(s_0; g) > 0$, i.e., the goal can be reached from the initial state.
2. For all $s \in \mathcal{S}$, $a \in \mathcal{A}$, $\exists\, s' \in \mathcal{S} : P(s'|s, a) = 1$, i.e., the MDP transitions deterministically.

We now consider the state-space trajectory $(s_i)_{i=0}^T$ which optimizes $V^*$ under the dynamics of the MDP, that is

$$s_{i+1} = \underset{s' \in \mathcal{T}(\{s_i\})}{\operatorname{argmax}} \ V^*(s'; g) \quad \forall\, 0 \le i < T. \tag{S9}$$

We will show that, for a certain finite $T$, $s_T$ achieves the goal, i.e., $R(s_T; g) = 1$ and $R(s_i; g) = 0 \ \forall\, 0 \le i < T$. We note that, since the episode terminates when the goal is achieved and $\gamma(s; g) = \gamma(1 - R(s; g))$, Bellman optimality implies that $R(s_T; g) = 1 \iff V^*(s_T; g) = 1$, as shown in Equation S6.

From Equation S6 we can then derive

$$
\begin{aligned}
V^*(s_i; g) &= R(s_i, g) + \gamma \max_{a \in \mathcal{A}} \underset{s' \sim \mathcal{P}(\cdot|a, s_i)}{\mathbb{E}} V^*(s'; g) \\
&= \gamma \max_{a \in \mathcal{A}} \underset{s' \sim \mathcal{P}(\cdot|a, s_i)}{\mathbb{E}} V^*(s'; g) \\
&= \gamma \max_{s' \in \mathcal{T}(\{s_i\})} V^*(s'; g) \\
&= \gamma V^*(s_{i+1}; g),
\end{aligned}
\tag{S10}
$$

where the third equality follows from deterministic transitions, and the last from the objective in Equation S9. By applying this equality recursively, we can write $V^*(s_i; g) = \gamma^{-i} V^*(s_0; g)$. After $T = \lceil \log_\gamma V^*(s_0; g) \rceil$ steps,

$$V^*(s_T; g) = \gamma^{-T} V^*(s_0; g) \ge 1. \tag{S11}$$

Since $V^*(\cdot; g) \le 1$, therefore $V^*(s_T; g) = 1$ and $s_T$ achieves the goal.

An actor network is trained to output

$$
\begin{aligned}
\pi(s_i; g) &= \underset{a \in \mathcal{A}}{\operatorname{argmax}} \ Q^*(s_i, a; g) \\
&= \underset{a \in \mathcal{A}}{\operatorname{argmax}} \ R(s_i; g) + \gamma \underset{s' \sim \mathcal{P}(\cdot|a, s_i)}{\mathbb{E}} V^*(s'; g) \\
&= \underset{a \in \mathcal{A}}{\operatorname{argmax}} \ \underset{s' \sim \mathcal{P}(\cdot|a, s_i)}{\mathbb{E}} V^*(s'; g)
\end{aligned}
\tag{S12}
$$

Under deterministic dynamics, the optimal policy is therefore producing the trajectory which optimizes the objective in Equation S9, and thus reaches the goal.

### A.3 Model-based Planning and $\mathcal{T}$-local Optima

Section 5 pinpoints $\mathcal{T}$-local optima as a class of estimation artifacts in learned value functions. Following from Section 6.1, we now present two properties arising from the interaction between model-based planning, and a suboptimal value landscape, depending on the depth of $\mathcal{T}$-local optima, and the planning horizon.

First, we show that, given a goal $g \in \mathcal{G}$, model-based planning with horizon $H$ from a $\mathcal{T}$-local optimum point $s^*$ of depth $k \ge H$ fails to achieve the goal. We make the assumptions presented in Section 3.1, and additionally assume:

1. $\hat{\mathcal{P}} = \mathcal{P}$, i.e., exact dynamics estimation.
2. Model-based planning optimizes its objective exactly (see Equation S13).

---

[4]On top of those listed in Section 3.1.

3. For every state $s \in \mathcal{S}$ and goal $g \in \mathcal{G}$, if $V(s; g) \geq \max_{s' \in \mathcal{T}(\{s\})} V(s'; g)$, then $\exists \, \bar{a} \in \mathcal{A} :$ $P(s|s, a) = 1$. This implies the existence of self-loops for a set of states including $\mathcal{T}$-local optimum points.

We recall from Section 6.1 that, in this setting, model-based planning optimizes the following objective:

$$a_0^{H-1} = \underset{a_0^{H-1} \in \mathcal{A}^H}{\mathrm{argmax}} \; V(s_H; g) \quad \text{with } s_{i+1} \sim \hat{P}(\cdot|s_i, a_i), \tag{S13}$$

We note that, if $s_0 = s^*$ is a $\mathcal{T}$-local optima, then by assumption (3) an action $\bar{a}$ is available such that $P(s^*|s^*, \bar{a}) = 1$. If $a_i = \bar{a}$ for $0 \leq i < H$, then the objective in Equation S13 simply evaluates to $V(s^*; g)$. Any other action sequence will reach a state within the $H$-step future state distribution $\mathcal{T}^H(\{s^*\})$, and will evaluate to a lower or equal value as per the definition of $\mathcal{T}$-local optima of depth $k$: $V(s^*; g) \geq \max_{s' \in \mathcal{T}^H(\{s^*\})} V(s'; g)$. If ties are broken favorably [5], an MPC model-based planning algorithm will therefore execute $\bar{a}$, which deterministically transitions to the $\mathcal{T}$-local optimum point at each step. Thus, the goal is never reached.

Second, we show that, if the planning horizon $H$ is greater than the depth of all $\mathcal{T}$-local optima over the state space $\mathcal{S}$, open-loop execution of the action sequence returned through model-based planning successfully optimizes the value function (i.e., reaches a state with a higher value).

This property holds under additional assumptions, namely:

4. For all $s \in \mathcal{S}$, $a \in \mathcal{A}$, $\exists \, s' \in \mathcal{S} : P(s'|s, a) = 1$, i.e., the MDP transitions deterministically.
5. The action sequence optimizing the objective for model-based planning (Equation S13) is executed in its entirety before replanning, i.e., open-loop execution of the plan.
6. $V(s_0; g) < \max_{s' \in \mathcal{S}} V(s'; g)$, i.e., the initial state does not maximize the value function.

Since there are no $\mathcal{T}$-local optima of depth $k \geq H$, then there must exist a state $s_1 \in \mathcal{T}^{H'}(\{s_0\})$ such that $V(s_1; g) > V(s_0; g)$ and $H' \leq H$. Now, let us consider the state reachable in up to $H$ steps with the highest value:

$$V(s_1; g) = \max_{H' \leq H} \max_{s' \in \mathcal{T}^{H'}(\{s_0\})} V(s'; g) \tag{S14}$$

As shown in Equation S13, model-based planning only optimizes over states that can be reached in exactly $H$ steps. It is thus sufficient to show that $s_1 \in \mathcal{T}^H(\{s_0\})$, as this would ensure that

$$V(s_0; g) < V(s_1; g) \leq \max_{s^H \in \mathcal{T}^H(\{s_0\})} V(s_H; g) = \max_{a_0^{H-1} \in \mathcal{A}^H} V(s_H; g) \text{ with } s_{i+1} \sim \hat{P}(\cdot|s_i, a_i). \tag{S15}$$

Thus, open-loop execution of the action sequence $a_0^{H-1}$ optimizing Equation S13 would reach a state with a higher value (in this case, $s_H$).

We recall that $s_1 \in \mathcal{T}^{H'}(\{s_0\})$ for some $H' \leq < H$. We will now show that $s_1 \in \mathcal{T}^H(\{s_0\})$ by contradiction. Let us assume that $s_1 \notin \mathcal{T}^H(\{s_0\})$. We thus have that

$$
\begin{aligned}
V(s_1; g) &= \max_{H' \leq H} \max_{s' \in \mathcal{T}^{H'}(\{s_0\})} V(s'; g) \\
&\geq \max_{H' < H'' \leq H} \max_{s' \in \mathcal{T}^{H''}(\{s_0\})} V(s'; g) \\
&\geq \max_{H' < H'' \leq H} \max_{s' \in \mathcal{T}^{H''-H'}(\{s_1\})} V(s'; g) \\
&\geq \max_{s' \in \mathcal{T}(\{s_1\})} V(s'; g)
\end{aligned}
\tag{S16}
$$

---

[5] Given two action sequences that both maximize the objective in Equation S13, we need to assume that the agent breaks the tie by maximizing the value of the first state in the trajectory $V(s_1; g)$. This assumption would not be necessary in case of open-loop execution of the action sequence.

Following from assumption (4), a self-loop is thus available. Therefore $s_1 \in \mathcal{T}(\{s_1\})$, and more importantly $s_1 \in \mathcal{T}(\{s_0\})^{H''}$ for any $H'' \geq H'$, thus contradicting the initial assumption and completing the proof.

# B  Occurrence of Artifacts and Downstream Performance

Sections 5 pinpoints $\mathcal{T}$-local optima as a class of estimation artifacts in learned value functions. This section empirically validates the harmfulness of such artifacts for an agent which attempts to optimize a value function $\hat{V}(s; g)$.

In order to do so, we train $n = 30$ value functions for each environment. We then optimize each of them through model-based planning with horizon $H$ for different goals, and obtain a set of trajectories. We split all evaluation trajectories in two groups: successes (those that achieve the goal and obtain a return greater than 0) and failures. We then estimate the occurrence of $\mathcal{T}$-local optima of depth $k \geq H$ along each trajectory, i.e., the ratio of states in the trajectory which are likely to be $\mathcal{T}$-local optimum points, and plot its distribution for each group.

In practice, evaluating whether a candidate state $s \in \mathcal{S}$ is a $\mathcal{T}$-local optima is unfeasible, as it requires the evaluation of maximum operators over continuous spaces. A sampling-based estimation of $\mathcal{T}$-local optima remains possible, as described in Section 5, but will necessarily introduce a significant amount of noise. Nonetheless, we present the results of this procedure in Figure S8 (identical to Figure 5 in the main paper).

We note that the distributions of occurrences present long tails. We observe a general tendency for successful trajectories to contain fewer estimated $\mathcal{T}$-local optima, as the median occurrence is zero in three environments, and in any case never greater than the median occurrence for unsuccessful trajectories. This tendency is unsurprisingly the strongest in maze_large, which was observed to be prone to the occurrence of global estimation artifacts with depth $k \geq H = 15$.

We additionally report an analysis of the same trajectories with an indirect procedure, which does not rely on sampling. Appendix A.3 argues that, under several assumptions, model-based planning with horizon $H$ from a $\mathcal{T}$-local optimum point of depth $k \geq H$ is incapable of reaching the goal, or any state with a larger value. The reverse does not necessarily hold: lack of value improvement does not imply the presence of artifacts, but might result from a suboptimal optimization procedure. Moreover, in practical settings, not all necessary assumptions hold. Nevertheless, it is still possible to compute the ratio of states $s_i$ in evaluation trajectories for which $\hat{V}(s_i; g) \geq \hat{V}(s_{i+H}; g)$: this metric should tend to zero in the absence of estimation artifacts in the value function $\hat{V}(\cdot; g)$, assuming that the action selection procedure optimizes it correctly. We report similar plots for this metric, which we refer to as occurrence of non-monotonicities, in Figure S9. We observe that the patterns occurring in Figure S8 arise more evidently, as non-monotonicities tend to occur less frequently in successful trajectories.

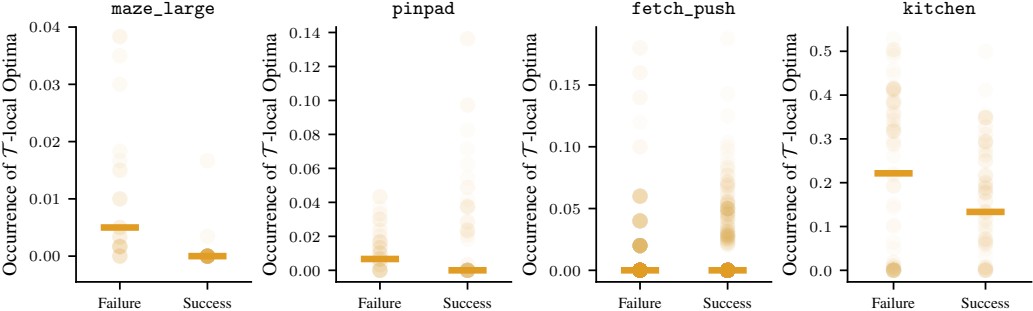

Figure S8: Occurrence of $\mathcal{T}$-local optima along successful and unsuccessful trajectories, estimated via sampling. An horizontal bar marks the median.

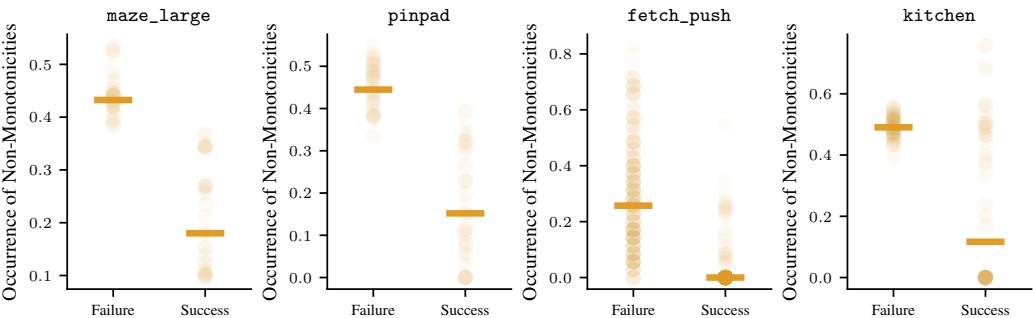

Figure S9: Occurrence of non-monotonicities along successful and unsuccessful trajectories. An horizontal bar marks the median.

## C    Comparison to Additional Reinforcement Learning Algorithms

In this Section we benchmark three additional value-learning methods for offline goal-conditioned reinforcement learning, namely IQL[22], LAPO [6] and WGCSL [51]. Additional hyperparameters are tuned with a limited budget on maze_large. Performances are reported for the original algorithms (actor network, in blue), and in combination with model-based planning (in orange). Results are averaged over 12 seeds and reported in Figure S10. The three additional methods perform within the range of those evaluated in the main paper, suggesting that they do not avoid value estimation artifacts. Moreover, they do not consistently outperform TD3, confirming the competitiveness of simple actor-critic algorithms. Finally, we observe that these additional methods generally improve their performance when coupled with model-based planning, as shown in Section 4.

## D    Comparison to Graph-based Methods

Discrete structures such as graphs have been explored in order to model long-horizon temporal distances over MDPs. Eysenbach et al. [12] have shown how shortest paths on a graph constructed by uniformly sampling a replay buffer can effectively be used to select subgoals in goal-conditioned RL, introducing Search on the Replay Buffer (SORB). Our graph-based value aggregation scheme generally follows this framework, but with several differences, which we find to be crucial when learning from exploration trajectories. In this section, we expand the method description from Section 6.2, and outline these differences.

The first, and perhaps most crucial, difference lies in how shortest paths are leveraged to guide a goal-conditioned agent. In Eysenbach et al. [12], shortest paths are only used to retrieve the second state along them, which is then commanded to the agent as a subgoal. In practice, we found this to potentially result in the selection of subgoals which are *too close*, and therefore cause myopic

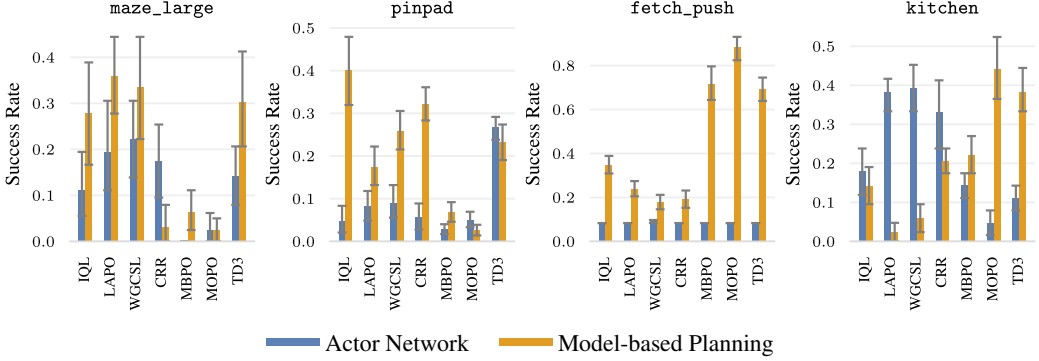

Figure S10: Extension of Figure 2 to additional value learning methods. We confirm that TD3 remains, on average, competitive with offline and model-based baselines when trained on curious exploration data.

behavior in the agent. For instance, in `maze_large`, the agent is encouraged to move at lower velocities to avoid overshooting its closest subgoal. While this could be addressed by conditioning the agent on subgoals further along the shortest path, this solution can be counterproductive, and force the agent to follow unreliable long-horizon value estimates. In contrast, we employ the entirety of the shortest path in order to compute long-horizon estimates by aggregating short-horizon estimates along this path. Conveniently, aggregates can be directly optimized through model-based planning, without needing to command intermediate subgoals, or to retrain the policy to match corrected value estimates.

A second difference with respect to existing works [12] lies in the sampling procedure for vertices in the graph. Instead of adopting uniform sampling, we sample states according to the inverse of a learned density estimate. In practice, we use an Exponential Kernel Density Estimator with bandwidth $h = 20$, where the distance between two states is simply the temporal distance $d(s, s') = \log_\gamma V(s; s')$. This makes the density estimates independent of the parameterization of the state space $\mathcal{S}$, thus avoiding the strong assumption that Euclidean distances in the state space are meaningful.

Third and last, once the graph has been constructed, a pruning threshold $V_{\min}$ is usually enforced to remove all edges encoding long-horizon distance estimates, which have low value estimates $V(s'; s'') < V_{\min}$. For this reason, in principle, this threshold should be as large as possible. However, an excessively high threshold would result in a disconnected graph, in which shortest paths may not exist. While this threshold is conventionally hand-tuned, we propose to select it automatically, by choosing the largest value $V_{\min}$ that ensures connectivity from the agent's state to the goal. This can be efficiently computed through a Dijkstra-like algorithm. We refer to our codebase for its implementation.

Having outlined three key differences with respect to previous methods, we now provide an empirical evaluation of their effects on downstream performance in Figure S11. All success rates reported are obtained by optimizing value functions learned through TD3 with model-based planning and default hyperparameters.

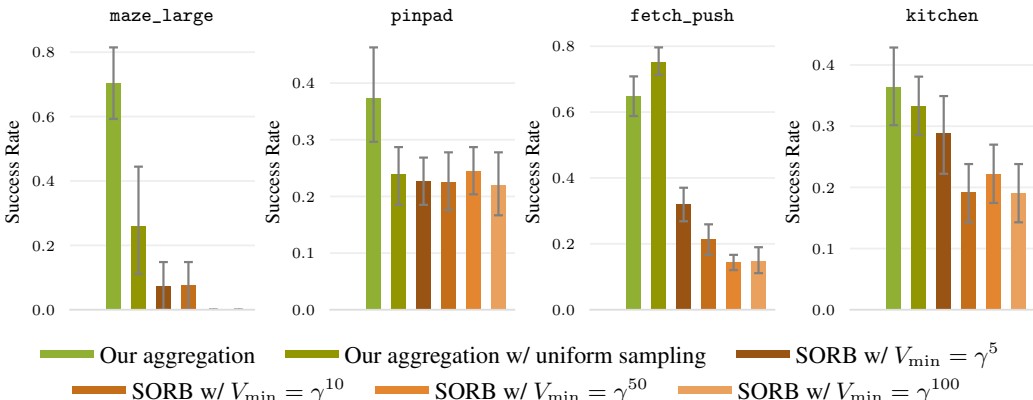

Figure S11: Ablation and comparison of graph-based methods. Our value aggregation scheme largely outperforms a naive application of SORB to this setting, while removing the need to tune the graph pruning threshold $V_{\min}$. We find that density-aware sampling of vertices in the graph is crucial for succeeding in long-horizon tasks.

In particular, we compare our graph-based value aggregation scheme (Our aggregation) with an ablation that samples graph vertices uniformly from the replay buffer. We observe that this results in a sharp decrease in performance in long-horizon tasks. We also replace our value aggregation scheme with a SORB-like subgoal selection procedure for different pruning thresholds $V_{\min}$. We find that a naive application of SORB to this setting is not effective, and can be detrimental with respect to simple model-based planning.

Removing the necessity to tune $V_{\min}$ leaves a single hyperparameter to be chosen for graph-based value aggregation, namely the size of the graph. This number is limited by the complexity of the shortest path search, and thus we use the default value of 1000 from the literature. However, we

found 100 vertices to already achieve sufficient coverage in `pinpad`, which therefore adopts this reduced hyperparameter.

In terms of limitations, as the graph aggregation procedure involves learned value estimates, it remains prone to failure due to compounding errors. However, two design choice introduce a degree of robustness. First, value estimates are only computed over states and goals that appear in the dataset the value function was trained on, which alleviates OOD issues. Second, the pruning step removes all edges encoding long-horizon estimates, which are empirically found to be less accurate [12]. Moreover, the tuning procedure we propose for the pruning threshold effectively minimizes the number of paths in the graph, while ensuring the existence of a solution to the optimization problem in Equation 5. Despite this, for particularly sparse datasets, the pruning can fail to remove inaccurate estimates, and the graph-based aggregates may reproduce the artifacts of the underlying value estimator. Further limitations of the graph search include increasing computational costs, and artifacts due to the effective discretization of the state space: while increasing the sample size can address the latter, it aggravates the former in a tradeoff.

# E    Sample Efficiency

All value functions in previous experiments are trained on 200k transitions gathered through curious exploration. This amount of data is around an order of magnitude lower than popular offline datasets [14], and has been chosen to be realistically collectable on real hardware. In practice, 200k timesteps at 30 Hz amount to little under 2 hours of real time interaction. Nonetheless, we also evaluate methods from Figure 7 (main paper) on smaller datasets, namely with 50k and 100k transitions. We visualize results in Figure S12, and find them to remain consistent, while success rates decrease on smaller datasets, as expected.

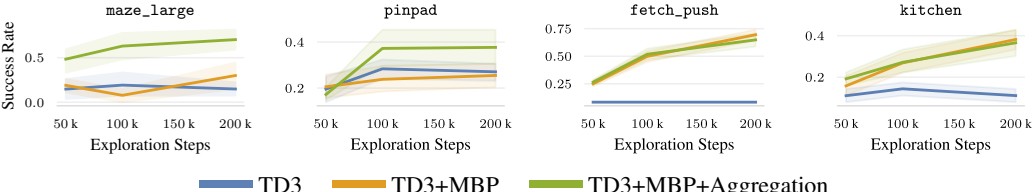

Figure S12: Performance of an actor network (TD3) against model-based planning without (TD3+MBP) and with graph-based value aggregation (TD3+MBP+Aggregation), reported for value functions trained on datasets with 50k, 100k, and 200k samples.

# F    Applicability to General Offline Goal-Conditioned Settings

The motivation of this work is to address the challenge of extracting goal-conditioned behavior after curious exploration. Focusing on data gathered through unsupervised exploration has value in itself, arguably, as this requires minimal supervision and can enable real-world robot learning [29]. However, the proposed method can in principle be applied to general offline goal-conditioned settings. This section explores this possibility and provides empirical evidence.

Throughout this work, the exploratory nature of the data distribution is leveraged mainly in two points. First, as argued in Section 4.2, when learning from curious exploration data, simple actor-critic methods are, on average, competitive with model-based or offline methods. In standard offline settings with near-expert data, offline algorithms should be expected to be more effective [14]. However, within the realm of curious exploration, Section 2 suggests that TD3 is equally performant, and thus the proposed method uses it as core algorithm for value learning. Second, the proposed method relies on model-based planning, whose performance is heavily dependent on the quality and robustness of the learned model. Model-based planning is thus particularly promising when learning from curious exploration, as the unsupervised exploration phase already produces a dynamics model, which does not need to be trained from scratch. Moreover, as data has been collected to maximize the information gain of this dynamics model, it is potentially more robust [38]. When learning from offline datasets, a model is generally not available: while it can also be learned offline, training it to high accuracy can be difficult.

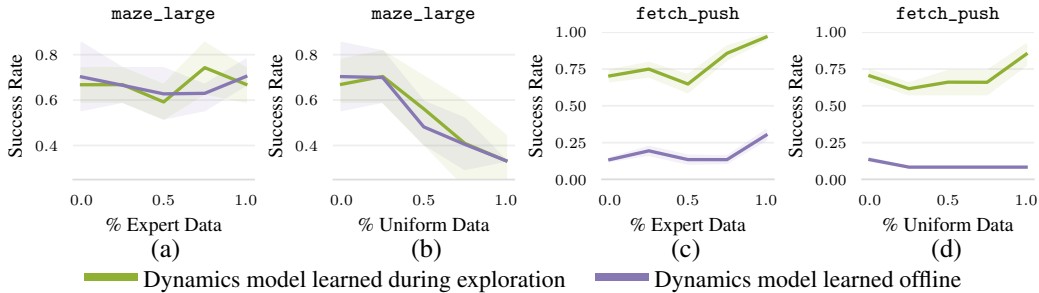

Figure S13: Ablation mixing the original dataset collected through curious exploration with expert or random uniform datasets. Our method can in practice be applied to different data sources, although may it suffer from weaker exploration in long-horizon environments (e.g. maze_large). Environments with more complex dynamics (e.g. fetch_push) largely benefit from the dynamics model learned during exploration (with active data collection).

We provide a minimal empirical study of the applicability of the proposed method on different data sources in Figure S13. Each subplot presents the performance of our full method (TD3 + model-based planning + graph-based value aggregation) when mixing the original curious exploration datasets with data collected from different sources (the size is fixed at 200k samples). In particular, a fraction of the trajectories is replaced with transitions from an expert policy (a,c), and from a uniformly random policy (b,d). Performance curves are recorded when using the dynamics model learned during curious exploration (green), and a dynamics model learned offline on the mixed data (violet); offline model learning was not tuned specifically. Due to time and compute constraints, we also present this analysis over two representative environments (maze_large in (a,b) and fetch_push in (c,d)), and average over 9 random seeds.

In these settings, we observe that access to expert data does not hinder success rates (Figure S13a,c): interestingly, theproposed method shows robustness to shrinking data support. On the other hand, when introducing data collected by a uniform random policy (Figure S13b,d), performance decreases accordingly to the reduced exploration: while collecting random trajectories is sufficient in fetch_push, it is not comparable to curiosity-driven exploration for long-horizon tasks (i.e., maze_large). Furthermore, we note that models trained offline (in violet) achieve generally success when the complexity of the environment's dynamics increases (fetch_push), confirming that online training via disagreement maximization returns a more robust model [38].

## G  Comparison to Value-driven Exploration

In this Section, we provide an informative comparison with a related framework: POLO [27]. POLO also combines value learning and model-based planning. However, there are fundamental differences:

- POLO explores by maximizing the uncertainty of the value function; our framework leverages the uncertainty of the dynamics model.
- POLO assumes the availability of ground truth models for planning; our framework does not and adopts a learned dynamic model.
- POLO does not rely on graph-based components for long-horizon planning.

While an official implementation of POLO is not available, we can loosely reproduce it by integrating POLO's exploration signal in our framework, replacing TD3 with its value learning scheme, and disabling graph-based value aggregation, while still relying on iCEM for model-based planning and learning a dynamics model. We name the resulting method POLO*. We find that POLO* suffers from model inaccuracy when computing its value targets via MPPI optimization. We thus introduce a variant which uses on-policy value targets, which we refer to as POLO**.

Figure S14 reports success rates for POLO* and POLO** against our method (TD3 + model-based planning + graph-based aggregation) in all of our benchmark environments. Plots are averaged over 12 seeds, except for POLO* which uses 5 due to higher computational cost. Moreover, we

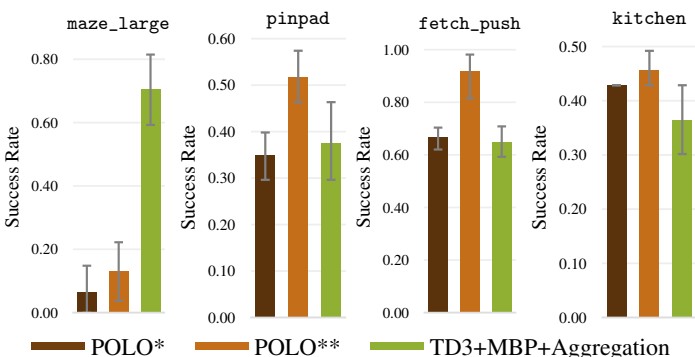

Figure S14: Comparison with adapted implementations of POLO – a privileged baseline that uses value-driven exploration and online value learning.

plot exploration trajectories in `maze_large` in Figure S15. We note that POLO is a privileged baseline, as it requires goals to be specified during training, and its value function is trained online by optimizing against its uncertainty. Our method, on the other hand, learns its value function offline from unlabeled data, and is therefore more widely applicable.

While the closer implementation (POLO*) is less performant, the improved version (POLO**) performs comparably or better than our method in three out of four environment. Despite successfully leveraging online value learning, we find its potential for long-horizon exploration to be limited, as shown by a large drop in performance in maze_large. We confirm this by visualizing trajectories gathered by POLO** (value-driven), and by the unsupervised exploration phase used in our work (dynamics-driven) in Figure S15.

## H    Compact Algorithmic Description

We report the full method (TD3 + MDP + Aggregation) in a compact algorithmic description in box S1, as well as in a short verbal summary below.

Our work deals with extracting a goal-reaching controller from the outcomes of curious exploration. Exploration data is collected while learning a dynamic model ensemble by maximizing its disagreement through model-based planning with zero-order trajectory optimization. This constitutes the intrinsic phase, which returns an exploration dataset, and a trained dynamics model. We then train a value function offline on the exploration dataset with TD3. During inference, we are given a goal. We select actions through model-based planning with the learned dynamics model. This time, actions

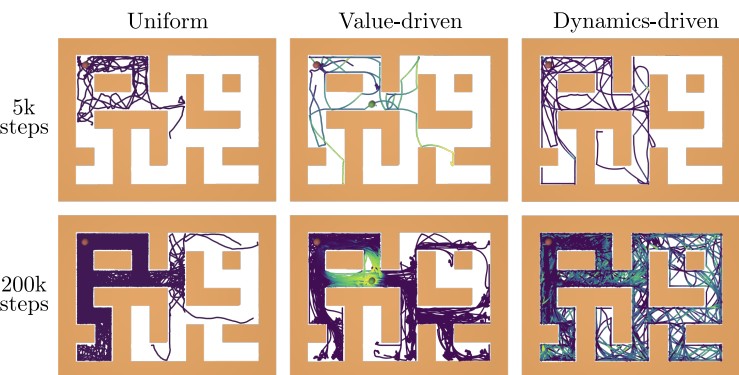

Figure S15: Random uniform, value-driven (as in POLO) and dynamics-driven (default in our method) exploration in `maze_large`. Lighter colors represent high exploration incentive. Uniform exploration is insufficient, and dynamics-driven exploration achieves dense coverage of the state space while being task-agnostic.

sequences are selected by maximizing the aggregated value of the last state in imagined trajectories, where aggregation is performed with graph search over the value estimates from TD3.

---

**Algorithm S1** TD3 + MDP + Aggregation

---
1: **for** each exploration episode **do**                                         ▷ Intrinsic Phase
2:     Collect a trajectory by maximizing the intrinsic reward $r_{\text{int}}$ (S17, S18) with iCEM and the dynamics model $\hat{\mathcal{P}}$.
3:     Add the trajectory to the replay buffer $\mathcal{D}$.
4:     Train the dynamics model $\hat{\mathcal{P}}$ via MLE over the replay buffer $\mathcal{D}$.
5: **end for**
6: **for** each offline training epoch **do**
7:     Train a goal-conditioned value function $\hat{V}$ with TD3 on the replay buffer $\mathcal{D}$, relabeled as described in Subsection 4.1.
8: **end for**
9: **for** each evaluation goal **do**                                             ▷ Extrinsic Phase
10:    Build graph **G** as described in Subsection 6.2.
11:    Act by maximizing the aggregated value $\bar{V}$ (5, S19) with iCEM and the dynamics model $\hat{\mathcal{P}}$.
12: **end for**

---

# I   Implementation Details

This section provides an in-depth description of methods and hyperparameters. Our codebase builds upon `mbrl-lib` [34], and adapts it to implement unsupervised exploration and goal-conditioned value-learning algorithms. To ensure reproducibility, we make it publicly available [6].

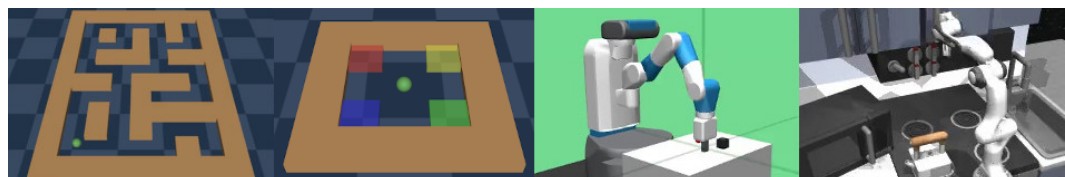

Figure S16: Environments used for evaluation: from left to right, `maze2d_large`, `pinpad`, `fetch_push` and `kitchen`.

## I.1   Environments

Empirical evaluations rely on four simulated environments. `maze_large` is adapted from D4RL [14], with the sole modification of fixing the initial position of the agent in a corner of the maze, in order to make exploration more challenging. The goal distribution includes 3 goals, placed on far-away corners of the maze, as displayed in Figure 6 (main paper). `pinpad` is implemented on top of D4RL [14] as a continuous, state-based version of the environment introduced in Hafner et al. [18]. The agent is initialized in a room with 4 buttons at its corners, and is tasked with pressing 3 of them in the correct order. Observations are thus 7 dimensional, and include 2D position and velocity of the agent, as well as a 3-dimensional history of the last buttons pressed. The agent is controlled through acceleration, as in `maze_large`. `kitchen` is adapted from gymnasium-robotics [13] in order to allow goal-conditioning. The goal distribution includes controlling each of the seven objects individually: a sliding door, a cabinet, a light switch, two burners, a teapot and a microwave oven. We use a frame skip value of 50 to encourage exploration. `fetch_push` is one of the benchmarks presented in Andrychowicz et al. [1], and we use it in its unmodified form, as provided in gymnasium [13]. A summary of further environment parameters can be found in Table S1.

---

[6]Code available at sites.google.com/view/gcopfce

|  | $\mathcal{|S|}$ | $\mathcal{|G|}$ | $\mathcal{|A|}$ | Episode Length | $\gamma$ |
|---|---|---|---|---|---|
| maze_large | 4 | 2 | 2 | 600 | 0.99 |
| pinpad | 7 | 3 | 2 | 300 | 0.99 |
| fetch_push | 25 | 3 | 4 | 50 | 0.95 |
| kitchen | 30 | 17 | 8 | 280 | 0.99 |

Table S1: Environment parameters.

## I.2 Unsupervised Exploration

Unsupervised exploration of the environment relies on curious model-based planning [38, 43]. We will thus describe the exploration algorithm, followed by the architecture of the dynamics model $\hat{\mathcal{P}}$ and the computation of the intrinsic cost signal.

Unsupervised exploration alternates environment interaction with model training, until sufficient experience is collected. We warm-start exploration by executing a uniformly random policy for 3000 steps, and saving its trajectories in the replay buffer $\mathcal{D}$. Then, before each episode, the dynamics model $\hat{\mathcal{P}}$ is trained on data from the replay buffer $\mathcal{D}$ via supervised learning for 12 epochs. Adam [21] is used as an optimizer with batch size $B = 512$, learning rate $\gamma_{\text{lr}} = 0.00028$ and weight decay parameter $\lambda = 0.0001$.

The dynamics model $\hat{\mathcal{P}}$ is learned as an ensemble of Gaussian neural networks, as suggested in Chua et al. [8]. Given a state $s \in \mathcal{S}$ and an action $a \in \mathcal{S}$, each of the $M$ ensemble member outputs a Gaussian distribution parameterized by its mean and its variance: $\{f_m(s, a) = [\mu_m(s, a), \text{Var}_m(s, a)]\}_{m=1}^M$. In practice, inputs and outputs are normalized, and the model predicts state differences. By default, the ensemble contains 7 members; during inference, only the best 5 members according to log-likelihood on a validation set are queried. Implementation and hyperparameters, reported in Table S2, are the default from mbrl-lib [34].

Exploration episodes are controlled through finite-horizon model-based planning with zero-order trajectory optimization (i.e., iCEM with the learned model $\hat{\mathcal{P}}$). Given a state-action pair $(s, a)$, an intrinsic reward signal can be computed as the trace of the covariance matrix over the ensemble's predictions [38, 49]

$$r_{\text{int}}(s, a) = \text{Tr}(\text{Cov}(\{f_m(s, a)\}_{m=1}^M)). \quad (S17)$$

This reward function can be aggregated into a cost function for trajectory optimization, as described in Section I.4. Overall, our implementation of unsupervised exploration runs at 5K steps per hour on a single Nvidia RTX3060 GPU or equivalent.

| Hyperparameter | Value |
|---|---|
| # of ensemble members | 7 |
| # of elites | 5 |
| # of layers | 4 |
| # of units per layer | 200 |
| Activation function | SiLU |
| Propagation method | TS1 [8] |
| # of particles | 20 |

Table S2: Hyperparameters for the dynamics model $\hat{\mathcal{P}}$.

## I.3 Value Learning

The relabeling procedure for exploration data is adapted from Tian et al. [46], and described in Section 4. We implement TD3 [14] and CRR [50] for value learning, and build upon TD3 for two model-based algorithms: MBPO [19] and MOPO [53]. We found DDPG [26] and SAC [17] to be slightly less performant in comparison to TD3. Each value learning algorithm is trained for 200 epochs, which, for a buffer size of 200K, corresponds to a wall-clock time less than 3 hours on a single NVIDIA RTX3060 GPU. Hyperparameters are listed in Table S3. Those tagged as (MBPO) are adopted by both MBPO and MOPO, while those tagged as (MOPO) or (CRR) are algorithm-specific. Remaining hyperparameters are shared by all algorithms. All hyperparameters are shared across environments, and they were tuned through a grid search across maze_large and fetch_push.

## I.4 Model-based Planning

Model-based planning is used both during the exploration phase, and during goal-conditioned evaluation. During exploration, the cost of each trajectory $(s_0, a_0, \ldots, s_{H-1}, a_{H-1})$ is computed as:

$$c_{\text{int}}((s_0, a_0, \ldots, s_{H-1}, a_{H-1})) = -\sum_{t=0}^{H-1} r_{\text{int}}(s_t, a_t) \tag{S18}$$

During goal-conditioned evaluation, a value function $V(s; g)$ is optimized, and the cost of each trajectory is directly adapted from the reinforcement learning objective:

$$
\begin{aligned}
c_{\text{ext}}((s_0, a_0, \ldots, s_{H-1}, a_{H-1})) &= -\left( \sum_{t=0}^{H-2} \gamma^t R(s_t; a_t) + \gamma^{H-1} V(s_{H-1}; g) \right) \\
&= -\gamma^{H-1} V(s_{H-1}; g),
\end{aligned}
\tag{S19}
$$

where the equality holds due to the extreme sparsity of $R(s; g) = \mathbf{1}_{s=g}$ in continuous spaces. Hyperparameters are fixed across environments, except for the planning horizon $H$. We find exploration to benefit from a longer horizon, which is thus set to $H = 30$. On the other hand, we found goal-conditioned evaluation to be more prone to model exploitation and hallucination. For this reason, during the extrinsic phase, we set $H = 10$ for `pinpad` and $H = 15$ for the remaining environments. For the same reason, we also compute the variance of particles for each trajectory, and discard those exceeding a set threshold.

While vanilla CEM [37] also performs well as zero-order trajectory optimizer, we found iCEM [35] to reach similar performances when using a reduced number of samples. Hyperparameters are reported in Table S4.

### I.5 Metrics

Unless specified, all plots show the mean estimate from 9 seeds, which are the combination of 3 seeds for collecting exploration data, and 3 seeds for value learning and planning for each dataset. Each mean estimate is accompanied by a 90% simple bootstrap confidence interval.

### I.6 Computational Costs

Our method's training costs (TD3 + model-based planning + aggregation) are those of TD3 ($\sim$70 minutes on a single NVIDIA RTX3060 GPU), comparable to other value-learning baselines ($\sim$50-90 minutes). Inference costs differ greatly: actor networks can be queried at >100Hz; model-based planning with and without aggregation at 3Hz and 0.3Hz respectively. An optimized implementation would bring significant speedups, e.g. through precomputation and parallelization of graph search, which could address the significant slowdown.

## J Open Challenges and Negative Results

Our method provides insights on how to mitigate the occurrence of estimation artifacts in learned value functions. However, it does not correct them by finetuning the neural value estimator. This possibility is interesting, as it would bring no computational overhead, and was broadly explored. TD finetuning with CQL-like penalties on the value of $\mathcal{T}$-local optima was attemped, but unsuccessful. In this context, we found such correction strategies to be sensible to hyperparameters, and often resulting in overly pessimistic estimates. We, however, stress the significance of direct value correction as an exciting direction for future research.

## K Numerical Results

Table S5 reports success rates from Figures 2 and 7 (main paper) in numerical form.

| Hyperparameter | Value |
|---|---|
| # of critic layers | 2 |
| # of units per critic layer | 512 |
| # of actor layers | 2 |
| # of units per actor layer | 512 |
| Batch size | 512 |
| Critic learning rate | 1e-5 |
| Actor learning rate | 1e-5 |
| Activation function | ReLU |
| Polyak coefficient | 0.995 |
| Target noise | 0.2 |
| Target noise clipping threshold | 0.5 |
| Policy delay | 2 |
| Policy Squashing | True |
| $p_g$ | 0.75 |
| $p_{\text{geo}}$ | 0.2 |
| (MBPO) # updates before rollouts | 500 |
| (MBPO) # of rollouts | 5000 |
| (MBPO) Rollout horizon | 5 |
| (MBPO) Buffer capacity | 1M |
| (MBPO) Real data ratio | 0.0 |
| (MOPO) Penalty coefficient $\lambda$ | 0.1 |
| (CRR) $\beta$ | 1.0 |
| (CRR) # of action samples | 4 |
| (CRR) Advantage type | mean |
| (CRR) Weight type | exponential |
| (CRR) Maximum weight | 20.0 |

Table S3: Hyperparameters for value learning.

| Hyperparameter | Value |
|---|---|
| # of iterations | 3 |
| Population size | 400 |
| Elite ratio | 0.01 |
| $\alpha$ | 0.1 |
| Population decay | False |
| Colored noise exponent | 3.0 |
| Fraction of kept elites | 0.3 |

Table S4: Hyperparameters for model-based planning.

| | maze_large | pinpad | fetch_push | kitchen |
|---|---|---|---|---|
| CRR | $0.095 < 0.173 < 0.253$ | $0.027 < 0.055 < 0.088$ | $0.083 < 0.083 < 0.083$ | $0.253 < 0.331 < 0.412$ |
| MBPO | $0.000 < 0.000 < 0.000$ | $0.017 < 0.028 < 0.040$ | $0.083 < 0.083 < 0.083$ | $0.111 < 0.143 < 0.174$ |
| MOPO | $0.023 < 0.049 < 0.079$ | $0.033 < 0.050 < 0.072$ | $0.083 < 0.083 < 0.083$ | $0.015 < 0.046 < 0.079$ |
| TD3 | $0.079 < 0.144 < 0.206$ | $0.244 < 0.267 < 0.291$ | $0.083 < 0.083 < 0.083$ | $0.079 < 0.110 < 0.142$ |
| CRR+MBP | $0.000 < 0.031 < 0.063$ | $0.283 < \mathbf{0.321} < 0.361$ | $0.152 < 0.190 < 0.231$ | $0.174 < 0.206 < 0.238$ |
| MBPO+MBP | $0.024 < 0.060 < 0.098$ | $0.043 < 0.068 < 0.091$ | $0.638 < 0.719 < 0.791$ | $0.174 < 0.222 < 0.285$ |
| MOPO+MBP | $0.000 < 0.025 < 0.098$ | $0.013 < 0.025 < 0.038$ | $0.824 < \mathbf{0.884} < 0.935$ | $0.365 < \mathbf{0.444} < 0.523$ |
| TD3+MBP | $0.206 < 0.301 < 0.412$ | $0.190 < 0.232 < 0.274$ | $0.638 < 0.693 < 0.745$ | $0.333 < \mathbf{0.379} < 0.428$ |
| TD3+MBP+Aggregation | $0.627 < \mathbf{0.705} < 0.814$ | $0.314 < \mathbf{0.410} < 0.500$ | $0.587 < 0.647 < 0.708$ | $0.301 < \mathbf{0.365} < 0.428$ |
| MBP+Sparse | 0.030 | 0.100 | 0.730 | 0.430 |

Table S5: Average success rates with 90% bootstrap confidence intervals from Figures 2 and 7 (main paper).

