# OpenReview forum: "Goal-conditioned Offline Planning from Curious Exploration"
_NeurIPS.cc/2023/Conference — NeurIPS 2023 poster_

### Official Review · Reviewer_yLbp · 2023-06-27

**Soundness:** 4 excellent
**Presentation:** 4 excellent
**Contribution:** 3 good
**Rating:** 6
**Confidence:** 4

**Summary:**

The paper presents a method that using a dataset of rollouts and a dynamic model create a graph like structure in-order to train goal conditioned policies. The nodes of the graph are selected based on it high local value function, and connected using a model based mechanism offline that avoid the myopic behavior of other methods.

**Strengths:**

Solid theoretical analysis of challenges on graph based models that create local and global minima.
In-depth comparison with other techniques, and prolific appendix with theoretical support, more results, and value function analysis.


**Weaknesses:**

Even if I see the intense amount of effort on the paper, its hard to piece all the contributions together. I'm mean more in the sense of clarity. One way of addressing this for example could be adding pseudocode of the algorithm.
The idea is an improvement of other graph based efforts of the past, as well pointed it out by the authors, Salinov et al , and others.
I believe is missing a comparison with the line of work of Lowrey et al 2019 ICRL, "https://sites.google.com/view/polo-mpc""Plan Online, Learn Offline: Efficient Learning and Exploration via Model-Based Control" , Hnsen et al 2022 , Neurips, and related work where a value function analysis it also performed, and the mixed of value function and model based planning is done.


**Questions:**

I would be interesting to see the comparison mentioned in the previous section.

---

> ### Author Rebuttal · Authors · 2023-08-09
>
> We thank the reviewer for their comments and suggestions, which we address sequentially as follows.
>
> # Contributions
> We have added to the Appendix a detailed algorithm box describing the practical algorithm, and how the different components are combined. In the context of this response, we would like to also report a verbal equivalent.
>
> Our work deals with extracting a goal-reaching controller from the outcomes of curious exploration. Exploration data is collected while learning a dynamic model ensemble by maximizing its disagreement through model-based planning with zero-order trajectory optimization. This constitutes the intrinsic phase, which returns an exploration dataset, and a trained dynamics model.
>
> We then train a value function offline on the exploration dataset with TD3. During inference, we are given a goal. We select actions through model-based planning with the learned dynamics model. This time, actions sequences are selected by maximizing the aggregated value of the last state in imagined trajectories, where aggregation is performed with graph search over the value estimates from TD3.
>
> We will reword the introduction to provide a similar description.
>
> # POLO
> We thank the reviewer for pointing us in the direction of POLO [1]. We believe that the method represents an interesting privileged baseline. The main differences between our method and POLO are as follows:
> - POLO explores by maximizing the uncertainty of the value function; our framework leverages the uncertainty of the dynamics model.
> - POLO assumes the availability of ground truth models for planning; our framework does not and adopts a learned dynamic model.
> - POLO does not rely on graph-based components for long-horizon planning.
>
> While an official implementation of POLO is not available, we can loosely reproduce it by integrating POLO's exploration signal in our framework, replacing TD3 with its value learning scheme, and disabling graph-based value aggregation, while still relying on iCEM for model-based planning and learning a dynamics model. We name the resulting method POLO*. We find that POLO* suffers from model inaccuracy when computing its value targets via MPPI optimization. We thus introduce a variant which uses on-policy value targets, which we refer to as POLO**.
>
> Figure B4 report success rates for POLO* and POLO** against our method (TD3 + model-based planning + graph-based aggregation) in all of our benchmark environments. Plots are averaged over 12 seeds, except for POLO* which uses 5 due to higher computational cost. Moreover, we plot exploration trajectories in maze_large in Figure B5.
> We note that POLO is a privileged baseline, as it requires goals to be specified during training, and its value function is trained online by optimizing against its uncertainty. Our method, on the other hand, learns its value function offline from unlabeled data, and is therefore more widely applicable.
>
> While the closer implementation (POLO*) is less performant, the improved version (POLO**) performs comparably or better than our method in three out of four environment. Despite successfully leveraging online value learning, we find its potential for long-horizon exploration to be limited, as shown by a large drop in performance in maze_large. We confirm this by visualizing trajectories gathered by POLO** (value-driven), and by the unsupervised exploration phase used in our work (dynamics-driven) in Figure B5.
>
> # References
> 1) Lowrey et al., Plan Online, Learn Offline: Efficient Learning and Exploration via Model-Based Control, ICLR 2019

---

### Official Review · Reviewer_nGpW · 2023-06-27

**Soundness:** 3 good
**Presentation:** 4 excellent
**Contribution:** 3 good
**Rating:** 6
**Confidence:** 4

**Summary:**

This paper tackles the problem of performing goal-conditioned control using data collected from curious exploration. They first perform an empirical evaluation of several methods for learning value functions from this offline data, and then identify an issue with estimated value functions called $\mathcal{T}$-local optima, and then a value aggregation scheme that grounds the value function in states observed during exploration, which essentially replaces a long-horizon value function estimate with a combination of shorter-horizon estimates. The paper finds that this aggregation strategy combined with model-predictive control is able to improve performance on tasks including maze solving, pinpad, fetch push, and Franka kitchen.

**Strengths:**

- The paper is well-motivated and clearly written. Each section was easy to follow and the hypotheses and experimental settings very sensible.
- The observation of $\mathcal{T}$-local optima as an estimation artifact and a cause of issues in goal-conditioned value function learning is very interesting and to my knowledge, has not been discussed before. While the authors’ empirical analysis in showing that $\mathcal{T}$-local optima indeed are the source of pathologies across multiple environments in the supplementary material is a bit less conclusive than Figure 5 would lead one to believe, I believe that this is a solid contribution that can help to direct future research to look into this further.
- The authors’ proposed solution to address these local optima via a combination of model-based planning and a (to my knowledge) novel graph aggregation scheme seems sensible and effective. It achieves improved (on maze_large and pinpad) or similar (on fetch_push and kitchen) performance.
- The empirical analysis of actor networks versus model-based planning helps to solidify some observations that have been made in prior work.

**Weaknesses:**

- I would recommend being more transparent with the results for an experiment like Figure 5 for the other environments, where the trend is not as apparent. This would be more helpful for readers to understand how well this hypothesis holds up empirically, and while I understand there is a significant amount of noise, that itself is also useful to know about.
- While the manuscript places significant emphasis on the unsupervised exploration/curiosity component of the problem setting, it doesn’t appear that any component of the method is actually specific to the curious exploration setting, nor does it leverage any particular properties of curious exploration. It seems like all of the arguments that the paper makes would be equally valid when using data from a random policy or an expert policy.

**Questions:**

- How does the choice of path length in equation 5 affect performance?
- As Equation 5 minimizes over a number of paths composed of learned value estimates, does this result in compounding errors and model exploitation as well?
- Following my comment in “weaknesses”, are there specific properties of the curious exploration setting that make this method more effective? Will the performance of this method change significantly if presented with data from a random policy or near-expert policy?

** Update after rebuttal : I have read the authors' rebuttals. They address my concerns reasonably well, and I have chosen to maintain my rating. **

**Limitations:**

The authors discuss some of the limitations to their methods, particularly the computational requirements compared to policy learning. I don’t see direct potential negative societal impact for this work, so I think the discussion is sufficient, but would appreciate a further discussion of failure cases or limitations particularly of their graph aggregation method.

---

> ### Author Rebuttal · Authors · 2023-08-09
>
> We thank the reviewer for their assessment and detailed feedback. In this response, we comment on Figure 5, on the generality of the method, and provide clarifications on graph-based aggregation.
>
> # Figure 5
> We appreciate the suggestion for extending Figure 5 to include other environments from the Appendix, and have applied this change.
> As the reviewer notes, these metrics are not designed to exactly quantify the existence of value artifacts, but only to estimate their presence and effects; they are sampled-based, and thus noisy by nature, especially in low-data regimes. We will add these remarks directly at the end of Section 5 to ensure clarity. Furthermore, we would like to encourage the development of more robust probing methods as a further direction of research.
>
> # Generality of the method
> We agree that our method can be applied beyond the curious exploration setting, and potentially learn from different data sources. Nevertheless, the empirical evaluation in Section 4.2 and certain components of our method are motivated by curious exploration. We agree that certain ideas and components can be effective in broader settings, and we thus perform an empirical investigation on the applicability of our method to data collected from an expert, or a random actor. We refer the reviewer to the common response for a broader discussion and experimental results.
>
> # Graph-based aggregation
> The choice of path length in Equation 5 is not necessary: all paths over the (pruned) graph are considered (as proposed in [1]), and the minimization is in practice solved through Dijkstra's algorithm. As the minimization is over products of learned value estimates, it is indeed prone to failure due to compounding errors. There are, however, two design choices that introduce robustness.
>
> First, value estimates are only computed over states and goals that appear in the dataset the value function was trained on, which alleviates OOD issues. Most importantly, the pruning step (see line 159 in the Appendix) effectively removes all edges encoding long-horizon estimates, which are empirically found to be less accurate [1]. Overall, the minimization is therefore solved over a smaller number of paths with respect to those existing in the fully-connected graph.
>
> Moreover, the tuning procedure we propose for the pruning threshold minimizes the number of candidate paths, while ensuring the existence of a solution. Despite this, for particularly sparse datasets, the pruning can fail to remove inaccurate estimates, and the graph-based aggregates may reproduce the artifacts of the underlying value estimator.
>
> Further limitations of the graph search include increasing computational costs, and artifacts due to the effective discretization of the state space: while increasing the sample size can address the latter, it aggravates the former in a tradeoff. We have added a dedicated discussion of failure cases and limitations, with a particular focus on graph-based aggregation to Appendix D, thus extending the existing discussion on the details of this component.
>
> # References
> 1) Eysenbach et al., Search on the Replay Buffer: Bridging Planning and RL, NeurIPS 2019

---

### Official Review · Reviewer_7YsN · 2023-06-29

**Soundness:** 4 excellent
**Presentation:** 2 fair
**Contribution:** 3 good
**Rating:** 5
**Confidence:** 3

**Summary:**

This paper investigates the issue of mitigating estimation artifacts in the value function of off-policy RL. The authors adopt an unsupervised scheme to collect exploratory data, which is subsequently used to train both off-policy and offline RL algorithms. In their analysis, the authors identify a geometric characteristic known as $\mathcal{T}$-local optima, which represents inaccurate value landscapes. To address this issue, the authors propose a combination of model-based planning and graph-based aggregation, resulting in enhanced performance on long-horizon goal-conditioned tasks.

**Strengths:**

* The finding in this paper that learned value landscapes can hinder achieving long-horizon goals is novel and interesting.
* The proposed method, which combines model-based planning with graph-based value aggregation, makes sense and is demonstrated to enhance the performance of vanilla TD3.
* The empirical evidence provided in Appendix C through visualization effectively demonstrates the presence of estimation artifacts in the value landscapes.

**Weaknesses:**

* I have concern regarding the performance of the proposed method, as it fails to attain comparable results with previously published works, such as [1][2] on Maze2D, [3][4] on FetchPush, and [5][6] on Kitchen.
* Furthermore, the planning method incurs higher training and testing costs than prior offline goal-conditioned works [4][6], yet its performance does not meet the same standards. I recommend that the authors check this issue by conducting a comprehensive comparison, and elucidate the advantages of the proposed method over other offline goal-conditioned RL baselines and model-based planning baselines.
* The literature review in this work is inadequate as it fails to cite and discuss important prior offline goal-conditioned works [4][6][7][8]. Notably, the work by [8], which involves the creation of a graph from offline dataset, is particularly relevant. Furthermore, this paper overlooks prior goal-conditioned methods [7][9][10] that employ model-based planning.
* The evaluation benchmark appears to be limited, especially two environments are simple point-based tasks. To conduct a comprehensive study, it is essential to incorporate additional benchmarks such as AntMaze in D4RL, as well as the environments FetchSlide, FetchPick, and FetchHandReach used in [4].
* The paper would benefit from an improvement in its presentation and organization. A significant portion of the paper is devoted to discussing value learning, leaving only Section 6.3 for the main experimental results. To address this, I recommend compressing or relocating part of Section 2, Section 3, Section 4, and Section 5, perhaps to the Appendix. Additionally, it would be helpful to move some of the interesting evidence from Appendix C to the main paper.



[1] Chen X, Ghadirzadeh A, Yu T, et al. Lapo: Latent-variable advantage-weighted policy optimization for offline reinforcement learning[J]. Advances in Neural Information Processing Systems, 2022, 35: 36902-36913.

[2] Janner M, Du Y, Tenenbaum J B, et al. Planning with diffusion for flexible behavior synthesis[J]. arXiv preprint arXiv:2205.09991, 2022.

[3] Charlesworth H, Montana G. Plangan: Model-based planning with sparse rewards and multiple goals[J]. Advances in Neural Information Processing Systems, 2020, 33: 8532-8542.

[4] Yang R, Lu Y, Li W, et al. Rethinking goal-conditioned supervised learning and its connection to offline rl[J]. arXiv preprint arXiv:2202.04478, 2022.

[5] Kostrikov I, Nair A, Levine S. Offline reinforcement learning with implicit q-learning[J]. arXiv preprint arXiv:2110.06169, 2021.

[6] Emmons S, Eysenbach B, Kostrikov I, et al. RvS: What is Essential for Offline RL via Supervised Learning?[J]. arXiv preprint arXiv:2112.10751, 2021.

[7] Li J, Tang C, Tomizuka M, et al. Hierarchical planning through goal-conditioned offline reinforcement learning[J]. IEEE Robotics and Automation Letters, 2022, 7(4): 10216-10223.

[8] Mezghani L, Sukhbaatar S, Bojanowski P, et al. Learning Goal-Conditioned Policies Offline with Self-Supervised Reward Shaping[J]. arXiv preprint arXiv:2301.02099, 2023.

[9] Nasiriany S, Pong V, Lin S, et al. Planning with goal-conditioned policies[J]. Advances in Neural Information Processing Systems, 2019, 32.

[10] Charlesworth H, Montana G. Plangan: Model-based planning with sparse rewards and multiple goals[J]. Advances in Neural Information Processing Systems, 2020, 33: 8532-8542.


**Questions:**

Overall, I appreciate the authors for identifying the problem and proposing a novel model-based planning method to address it. I would like to raise my score if the authors can address the following concerns mentioned in the above section on weaknesses.
* The performance is not comparable to that of prior works in several environments.
* It is recommended to compare the results with additional baselines and report the computational costs for both training and testing.
* Please consider supplementing the literature review of related works.
* It would be beneficial to provide additional evidence from other benchmarks, such as AntMaze or FetchSlide/Pick/HandReach.
* Additionally, improving the presentation and organization of the paper would be helpful.
* Minor: Line 125 appears to contain a typo where $p_{G}$ is mentioned.


**Limitations:**

This paper highlights one limitation: the method does not address estimation artifacts during training. However, another significant limitation is the negative impact imposed by the computational costs associated with training the ensemble model, graph building, and test-time planning.

---

> ### Author Rebuttal · Authors · 2023-08-09
>
> We are grateful for the detailed comments and clear listing of points, which we address sequentially.
> Please let us know if further clarification or evidence is needed.
>
> # Performance comparison
> As noted by the reviewer, performance in the four environments is not comparable to that reported in previous work. The main reason is that our experiments do not use off-the-shelf offline datasets (e.g., D4RL [1]): we deliberately focus on datasets collected with minimal supervision and in realistic timeframes (see Section 3.2). The datasets are collected through curious exploration (using model disagreement as intrinsic motivation), which is fundamentally different from near-expert demonstrations, or uninformed exploration contained in standard datasets. While curious exploration attains better exploration compared to uniformly collected datasets [2], it does not generally contain high-quality, task-specific behavior (unlike most D4RL datasets).
>
> This naturally impacts downstream task performance. For instance, on maze_large, unsupervised exploration manages to visit all sections of the maze, albeit hard-to-reach ones are visited less often compared to the D4RL dataset (see Fig. 3), and we observe success rates above 70% (Figure 6). In more complex environments, we find curious exploration to be less performant. For instance, in kitchen, 2/7 objects are not interacted with during exploration, and downstream agents fail to control them.
>
> Data quantity is also limited: we focus on datasets that could be collected within a few hours at 30 Hz, totaling 200K transitions. Compared to the literature, our datasets are up to an order of magnitude smaller [1,3]. Our evaluation also slightly differs from the D4RL protocol: we assess multiple goals per environment, reporting success rates for interpretability, not returns (see Appendix E).
>
> For these reasons, we include multiple baselines, to which we added those suggested by the reviewer.
>
> # Additional baselines
> We implemented and benchmarked 3 of the baselines suggested, namely IQL, LAPO and WGCSL. We have prioritized them over the three remaining works, which are not value-based and are not reported to significantly outperform the three implemented ones. Additional hyperparameters were tuned with a limited budget on maze_large. We ran each baseline in its original configuration (actor network, in blue), and in combination with model-based planning (in orange). Performance is averaged over 12 seeds and reported in Fig. B2. The three baselines perform within the range of existing ones, suggesting that they do not avoid value estimation artifacts. Moreover, they do not consistently outperform TD3, confirming the competitiveness of simple actor-critic algorithms, as we hypothesized.
> Finally, the new baselines generally improve their performance when coupled with model-based planning, as shown for the existing ones.
>
> # Computational costs
> We have extended Appendix E.4 to include more precise computational costs, including baselines. Our method's training costs (TD3 + model-based planning + aggregation) are those of TD3 (\~70 minutes on RTX3060), comparable to other value-learning baselines (\~50-90 minutes). Inference costs differ greatly: actor networks can be queried at >100Hz; model-based planning with and without aggregation at \~3Hz and \~0.3Hz respectively (Appendix E.4). An optimized implementation would bring significant speedups, e.g. through precomputation and parallelization of graph search. The intrinsic phase also incurs computational costs (Appendix E.2).
>
> # Literature review
> We thank the reviewer for the pointers to relevant literature. We will rework and expand our related work section to include informative context on goal-conditioned reinforcement learning. We agree that [4] is particularly relevant, and it was in fact already included (see line 115), albeit the reference points to a different venue.
>
> # Additional Environments
> As suggested by the reviewer, we have applied our method to the four proposed environments.
> - antmaze is an interesting benchmark, involving low-level control, and high-level planning. However, the performance of our method, as well as baselines, is constrained by the unsupervised exploration phase. While being a state-of-the-art exploration signal, ensemble disagreement of the dynamics model is not able to make significant progress in this environment: the exploration algorithm focuses on exploring the dynamics of the robot, and does not explore beyond the initial corridor in the first 200k steps. All baselines, as well as our method, do not make progress in this task.
> - hand_reach is also constrained by the exploration data: the chaotic nature of disagreement-based curiosity results in not observing fine-grained motor control. As a result, we find that no method reaches the precision required by the original benchmark (~5 mm average error in fingertip positions). However, both our full method and a simple TD3 policy are able to solve the task for larger tolerances. We opt not to include it to avoid confusion.
> - we found exploration to be sufficient in fetch_pickandplace and fetch_slide. We thus extended the plots in Fig. 6 by Fig. B3. The results are consistent with those observed in fetch_push: model-based planning is crucial, while graph-based aggregation is not as beneficial, as any goal can already be reached within the planning horizon.
>
> # Presentation
> We have spatially compressed the initial discussion and added one example from Appendix C to the main paper to more clearly demonstrate learning artifacts, as suggested.
>
> # References
> 1) Fu et al., D4RL: Datasets for Deep Data-Driven RL, arXiv:2004.07219
> 2) Lampert et al., The Challenges of Exploration for Offline RL, arXiv:2201.11861
> 3) Yang et al. Rethinking Goal-conditioned Supervised Learning and its Connection to Offline RL, arXiv:2202.04478
> 4) Mezghani et al., Learning Goal-Conditioned Policies Offline with Self-Supervised Reward Shaping, CoRL 2022

---

> > ### Comment · Reviewer_7YsN · 2023-08-11
> > **Thanks for your response**
> >
> > Thank you for providing such a detailed response. I appreciate the inclusion of additional baselines and environments, as they enhance the empirical contribution of this paper. After reviewing the supplemented PDF, I am unclear about why many methods outperform the model-based planning approach in Figure B2.
> >
> > Furthermore, considering the reported computational cost, it appears that the method proposed in this paper requires a substantial amount of inference time (30$\times$ or 300 $\times$ slower than TD3). This could potentially be a major limitation of this work. Nevertheless, since the authors have addressed my other concerns, I am inclined to increase my score to 5.

---

> > > ### Author Response · Authors · 2023-08-13
> > >
> > > We thank the reviewer for the fast response, and we are glad that the inclusions were found to enhance the empirical contributions. We would like to briefly provide some further clarifications.
> > >
> > > Figure B2 suggests that model-based planning can, on average, improve performance with respect to actor networks, when learning from curious exploration data. We argue that this can be traced back to the ability of look-ahead planning to "escape" value artifacts (see Section 6.1). In general, planning with an (inaccurate) model can however also introduce issues, such as model exploitation (although, in this case the same model is shared by all baselines). Moreover, due to its multi-step optimization, the planner can potentially find distant states with a hallucinated high value that would not be found by one-step greedy maximization (i.e. actor networks). While we found this to be a minor issue in practice, it does appear to occur for certain baselines (e.g. offline model-free algorithms in kitchen, see Fig. B2).
> > >
> > > Finally, we would like to further comment on runtime. While the inference times reported for our implementation cannot be considered real-time, we are hopeful that this limitation can be overcome in the future. One option for improvement is code optimization, as our codebase was not designed for speed. Another option is hardware scaling. As a proof of concept, we benchmarked our implementation of the algorithms on stronger hardware (i.e. NVIDIA A100). We observed a significant speedup: our method can run a 8Hz and 2Hz, without and with graph-based aggregation respectively.
> > > We find this to be promising for two reasons. First, the speedups are significant (2.6x and 6.6x, respectively), and bring the method in line with others relying on model-based planning with learned models [1]. Second, we observe that the slowdown due to graph-based aggregation significantly reduces on better hardware (from 10x to 4x). In fact, the only difference implementation-wise when enabling graph-based aggregation is in the evaluation of the final states of imagined trajectories during planning. When graph-based aggregation is not used, this is done through a single forward pass of the critic. Interestingly, when aggregation is used, this can still be done with a single forward pass, albeit for a much larger batch, as pre-aggregation values must not only be conditioned on the environment's goal, but also on goals embedded in the vertices of the graph. The remaining estimates needed for graph search (e.g. pairwise values on the graph) can be precomputed. This suggests that, with improved capacity for hardware parallelism, the slowdown procured by graph-based aggregation would fade.
> > >
> > > We are happy to engage in any further discussion or clarification on this and other topics.
> > >
> > > ### References
> > > 1) Thodoroff et al. Benchmarking Real-Time Reinforcement Learning, NeurIPS 2021 Pre-registration Workshop

---

### Official Review · Reviewer_cdVD · 2023-07-07

**Soundness:** 3 good
**Presentation:** 3 good
**Contribution:** 3 good
**Rating:** 6
**Confidence:** 3

**Summary:**

This paper addresses the challenge of extracting goal-conditioned behavior without the need for continued environment interaction. The authors tackle this challenge by leveraging unsupervised exploration to collect diverse experiences. The study explores the suitability of different off-policy actor-critic algorithms and investigates the performance gap between utilizing network-based and model-based planning approaches. To overcome global value estimation artifacts, the authors propose a graph-based value aggregation method that aggregates short-horizon estimates. This technique aims to correct potential inaccuracies in global value estimation. The proposed framework is evaluated using various simulated environments.


**Strengths:**

- Analysis of the performance gap between model-based planning and actor networks is interesting.
- Extensive experiments


**Weaknesses:**

One notable weakness of this paper is the limited technical contribution. The two main components proposed, model-based planning and graph-based aggregation, are existing algorithmic components commonly used in the field. As a result, the paper lacks novelty in terms of introducing new techniques or approaches.

Furthermore, the proposed framework of using model-based planning and graph-based aggregation does not appear to be specifically tailored to the setup of achieving goals after an unsupervised exploration phase without further interaction with the environment. It would be beneficial to clarify whether certain components or analyses of the proposed framework are specifically relevant to this setup. If not, it might be more appropriate to broaden the scope to a generic goal-conditioned RL setup. Additionally, it would be valuable to see an evaluation of the proposed framework in the context of this broader setup.


**Questions:**

- in section 4.1, why do we need sampling negative goals?


**Limitations:**

The authors adequately addressed the limitations and potential negative societal impact.

---

> ### Author Rebuttal · Authors · 2023-08-09
>
> We thank the reviewer for their detailed feedback and clear assessment. In this response we further clarify our technical contributions, provide a discussion and experiments regarding the generality of our method and answer the reviewer's question.
>
> # Technical contributions
> While both model-based planning and graph-based techniques have been explored in the literature, we are not aware of previous works that analyze them under the common lens of robustness to local minima, or propose their combination. In particular, graph search has been mostly leveraged for subgoal selection [1] or policy distillation [2], and not embedded in the planning procedure.
>
> Moreover, from a strictly technical perspective, we propose significant modifications to the graph search procedure introduced in [1], as we find that a naive combination of existing methods is not effective. Namely, we propose importance sampling with learned density estimates for the vertices of the graph, and automatic threshold selection for graph pruning. This guarantees that the graph will be pruned as aggressively as possible, thus reducing reliance on inaccurate long-horizon estimates, while ensuring the existence of a solution to the graph search problem. A detailed discussion and ablation of these features can be found in Appendix D.
>
> # Applicability to general offline GC RL
> We agree that the scope of the method is not strictly limited to learning from curious exploration data. In fact, it can in principle be applied to general goal-conditioned settings. However, certain components of our analysis and method are specific to curious exploration or are designed to leverage the properties of exploratory data. We thank the reviewer for encouraging the exploration of this direction and refer them to the common response for a discussion and new experimental results.
>
> # Sampling of negative goals
> Finally, we would like to address the question about Section 4.1. If negative goals were not sampled, the value function would only be trained on goals that were actually reached from each given state. On the one hand, this would hinder trajectory stitching. On the other hand, the value function would systematically overestimate values, as any goal which is not easily achievable from a given state would never be observed during training. At inference, when a challenging goal is set, the value network would then be queried on a state-goal pair that is significantly different from those observed during training. Empirically, this results in a drop in performance when the probability of sampling negative (random) goals is set to zero.
>
> # References
> 1) Eysenbach et al., Search on the Replay Buffer: Bridging Planning and Reinforcement Learning, NeurIPS 2019
> 2) Mezghani et al., Learning Goal-Conditioned Policies Offline with Self-Supervised Reward Shaping, CoRL 2022

---

### Author Rebuttal · Authors · 2023-08-09

We thank all reviewers for their constructive feedback, which helped us to improve our paper. We ran additional baselines, added evaluations on more environments, and provided individual responses.
Here, we summarize the added validation, and then address the applicability of our method in general offline goal-conditioned settings, with arbitrary data sources.
We look forward to further discussing our work and providing clarifications or additional results.

# Summary
We gathered additional results which confirm our contributions, and provide further insights into our method.
- We add a study on the applicability of our method to general offline goal-conditioned settings (see Fig. B1 in the rebuttal PDF, and discussion below).
- We extend the experimental results on value estimation of Section 4.2 by introducing 3 additional baselines, i.e., IQL [4], LAPO [5] and WGCSL [6]. These methods perform similarly to benchmarked offline RL algorithms, thus confirming the effectiveness of standard methods such as TD3 in this setting. (see Fig. B2 and response to Reviewer 7YsN)
- We evaluate our method on two additional environments: fetch_pickandplace and fetch_slide. Results are consistent with those reported in fetch_push. (see Fig. B3 and response to Reviewer 7YsN)
- We study POLO [7] as an alternative (but not task-agnostic) value-driven exploration algorithm. We had to adapt the method as it assumes ground-truth models. By training its value function online, POLO achieves good performance in most environments. Our method, despite training its value function completely offline, is often able to match its performance and surpasses it when long-horizon exploration is necessary. (see Fig. B4 and response to Reviewer yLbp)

# Applicability to general offline goal-conditioned settings
The motivation of our work is to address the challenge of extracting goal-conditioned behavior after curious exploration. We believe that focusing on data gathered through unsupervised exploration has value in itself, as this requires minimal supervision and can enable real-world robot learning [1]. However, as reviewers noted, our method can in principle be applied to general offline goal-conditioned settings. In this section, we comment on this possibility and provide empirical evidence.

## Specificity to unsupervised curious exploration
Throughout this work, the nature of the data distribution is leveraged mainly in two points.
- As argued in Section 4.2, we find that, when learning from curious exploration data, simple actor-critic methods are, on average, competitive with model-based or offline methods. In standard offline settings with near-expert data, we expect offline algorithms to be more effective [2]. However, within the realm of curious exploration, we found TD3 to be equally performant, and thus use it as core algorithm for value learning.
- Our method relies on model-based planning, whose performance is heavily dependent on the quality and robustness of the learned model. Model-based planning is thus particularly promising when learning from curious exploration, as the unsupervised exploration phase already produces a dynamics model, which does not need to be trained from scratch. Moreover, as data has been collected to maximize the information gain of this dynamics model, it is potentially more robust [3]. When learning from offline datasets, a model is generally not available: while it can also be learned offline, training it to high accuracy can be difficult.

## Evaluation on different data sources
We provide an empirical study of the applicability of our method on different data sources in Fig. B1. Each subplot presents the performance of our full method (TD3 + model-based planning + graph-based value aggregation) when mixing the original curious exploration datasets with data collected from different sources (the size is fixed at 200k samples). In particular, we replace a fraction of the trajectories with transitions from an expert policy (a,c), and from a uniformly random policy (b,d). We include performance curves when using the dynamics model learned during curious exploration (green), and a dynamics model learned offline on the mixed data (violet); offline model learning was not tuned specifically. Due to time and compute constraints, we also present this analysis over two representative environments (maze_large in (a,b) and fetch_push in (c,d), and average over 9 random seeds.

In these settings, we observe that access to expert data does not hinder success rates (Fig. B1a,c): interestingly, our method shows robustness to shrinking data support. On the other hand, when introducing data collected by a uniform random policy (Fig. B1b,d), performance decreases accordingly to the reduced exploration: while collecting random trajectories is sufficient in fetch_push, it is not comparable to curiosity-driven exploration for long-horizon tasks (i.e. maze_large). Furthermore, we note that models trained offline (in violet) achieve generally success when the complexity of the environment's dynamics increases (fetch_push), confirming that online training via disagreement maximization returns a more robust model [3].

We have added these results to the Appendix and will refer to them in Section 4 to provide more context.

# References
1) Mendonca et al., ALAN: Autonomously Exploring Robotic Agents in the Real World, ICRA 2023
2) Fu et al., D4RL: Datasets for Deep Data-Driven RL, arXiv:2004.07219
3) Sancaktar et al., Curious Exploration via Structured World Models Yields Zero-Shot Object Manipulation, NeurIPS 2022
4) Kostrikov et al., Offline RL with Implicit Q-Learning, ICLR 2022
5) Chen et al., Lapo: Latent-variable Advantage-weighted Policy Optimization for Offline RL, NeurIPS 2022
6) Yang et al., Rethinking Goal-conditioned Supervised Learning and its Connection to Offline RL, ICLR 2022
7) Lowrey et al., Plan Online, Learn Offline: Efficient Learning and Exploration via Model-Based Control, ICLR 2019

---

### Decision · Program_Chairs · 2023-09-21

**Decision:**

Accept (poster)

**Comment:**

The paper tackles an important problem of extracting goal-conditioned policies from data collected by using curioisty-driven explorarion. Reviewers unanimously but mildly vote to accept the paper, I concur. Authors should consider including responses to reviewer questions and additional experiments in the final version of the paper.